# The effect of protecting women against economic shocks to fight HIV in Cameroon, Africa: The POWER randomised controlled trial

**Aurélia Lépine**[1]*, **Sandie Szawlowski**[1], **Emile Nitcheu**[2], **Henry Cust**[1], **Eric Defo Tamgno**[2], **Julienne Noo**[2], **Fanny Procureur**[1], **Illiasou Mfochive**[2], **Serge Billong**[3], **Ubald Tamoufe**[2]

**1** University College London, Institute for Global Health, London, United Kingdom, **2** John Hopkins Cameroon Program, Yaounde, Cameroon, **3** University of Yaounde I, Yaounde, Cameroon

* a.lepine@ucl.ac.uk

## Abstract

### Background

Women in sub-Saharan Africa are disproportionately affected by the HIV epidemic. Young women are twice as likely to be living with HIV as men of the same age and account for 64% of new HIV infections among young people. Many studies suggest that financial needs, alongside biological susceptibility, are a leading cause of the gender disparity in HIV acquisition. New robust evidence suggests women adopt risky sexual behaviours to cope with economic shocks, the sudden decreases in household's income or consumption power, enhancing our understanding of the link between poverty and HIV. We investigated if health insurance protects against economic shocks, reducing the need for vulnerable women to engage in risky sexual behaviours and reducing HIV and sexually transmitted infection (STI) incidence.

### Method and findings

We conducted a randomised controlled trial to test the effectiveness of a formal shock coping strategy to prevent HIV among women at high risk of HIV (registration number: ISRCTN 22516548). Between June and August 2021, we recruited 1,508 adolescent girls and women over age 15 years who were involved in transactional sex (*n* = 753) or commercial sex (*n* = 755), using snowball sampling. Participants were randomly assigned (1:1) to receive free health insurance for themselves and their economic dependents for 12 months either at the beginning of the study (intervention; *n* = 579; commercial sex *n* = 289, transactional sex *n* = 290) from November 2021 or at the end of the study 12 months later (control; *n* = 568; commercial sex *n* = 290, transactional sex *n* = 278). We collected data on socioeconomic characteristics of participants. Primary outcomes included incidence of HIV and STIs and were measured at baseline, 6 months after randomisation, and 12 months after randomisation. We found that study participants who engaged in transactional sex and were assigned to the intervention group were less likely to become infected with HIV post-

**Data Availability Statement:** All relevant data are within the manuscript and its Supporting Information files.

**Funding:** UKRI future leaders fellowship MR/S031790/1 awarded to Aurelia Lepine (AL). The funders had no role in study design, data collection and analysis, decision to publish, or preparation of the manuscript.

**Competing interests:** The authors have declared that no competing interests exist.

**Abbreviations:** AOR, adjusted odds ratio; ART, antiretroviral therapy; CBO, community-based organisation; CI, confidence interval; ELISA, zyme-linked immunosorbent assay; ITT, intention-to-treat; ME, marginal effect; OLS, ordinary least squares; OR, odds ratio; STI, sexually transmitted infection.

intervention (combined result of 6 months post-intervention or 12 months post-intervention, depending on the follow-up data available; odds ratio (OR) = 0.109 (95% confidence interval (CI) [0.014, 0.870]); $p$ = 0.036). There was no evidence of a reduction in HIV incidence among women and girls involved in commercial sex. There was also no effect on STI acquisition among both strata of high-risk sexual activity. The main limitations of this study were the challenges of collecting reliable STI incidence data and the low incidence of HIV in women and girls involved in commercial sex, which might have prevented detection of study effects.

## Conclusion

The study provides to our knowledge the first evidence of the effectiveness of a formal shock coping strategy for HIV prevention among women who engage in transactional sex in Africa, reinforcing the importance of structural interventions to prevent HIV.

## Trial registration

The trial was registered with the ISRCTN Registry: ISRCTN 22516548. Registered on 31 July 2021.

## Author summary

### Why was this study done?

- Women in Africa, especially adolescent girls and young women, suffer a greater burden of new HIV infections than men.

- Evidence suggests that women may engage in risky sexual behaviours, including unprotected sex, to earn extra money when faced with high expenses or income loss, also known as economic shocks.

- One of the most common economic shock in Africa is medical out of pocket payments, which can be dramatically reduced with health insurance.

### What did the researchers do and find?

- We conducted a randomised control trial assessing the impact of health insurance as a way of protecting women engaging in commercial sex or transactional sex from economic shocks to prevent HIV infection.

- We recruited 1,508 female participants, stratified depending on whether the participant is a female sex worker ($n$ = 755) or engage in transactional sex ($n$ = 753). Participants were randomly assigned (1:1) to receive free health insurance for themselves and their economic dependents for 12 months either at the beginning of the study (intervention; $n$ = 579) from November 2021 or at the end of the study 12 months later (control; $n$ = 568).

- We found that health insurance almost fully eliminated out-of-pocket medical spending and led to a small increase in health care use.

- Among women who received the intervention, we found large reductions in HIV infection and reductions in risky sexual behaviours for women and girls engaging in transactional sex, but there was no effect in women and girls engaging in commercial sex.

## What do these findings mean?

- These findings showed that preventing economic shocks among women engaging in transactional sex in Africa is an effective HIV prevention strategy.

- The results also highlighted that women in transactional sex were more affected by HIV than female sex workers in our study, a finding that warrants attention from policy makers.

- The main limitation of this study was the poor sensitivity and specificity of sexually transmitted infection tests.

## 1. Introduction

Women aged 15 to 24 years in sub-Saharan Africa are disproportionally affected by the HIV epidemic: they are twice as likely to live with HIV than men of the same age and account for 63% of new HIV infections among young people [1]. Today, new HIV infections are 3 times higher in young women than in their male counterparts [2]. In Cameroon, a country with one of the highest gender disparities in HIV globally, adolescent girls aged 15 to 19 years are 5 times more likely to be infected with HIV than their male counterparts (2% versus 0.4%) [3]. Globally, female sex workers are a key population in the fight against HIV and are now 26 times more likely to be infected than the general population, up from 13 times in 2017 [1,4]. There is growing evidence that heterosexual sex as part of commercial and transactional relationships (i.e., noncommercial sexual relationships in exchange for material support and benefits) are a key driver of the HIV epidemic in sub-Saharan Africa [5,6].

Many studies suggest that financial needs, alongside biological susceptibility, are the leading causes of the gender disparity in HIV acquisition through the adoption of risky sexual behaviours by young women associated with commercial and transactional sex [7–9]. While the lives of women and girls have dramatically improved over the last quarter century, there has been limited progress towards gender equality for the world's poorest, particularly among marginalised women or women living in remote areas [10]. Structural gender disparities mean women lack the same access as men to formal well-paid jobs, education, and productive assets (e.g., land), limiting their economic security [11]. This reduces the ability of women, particularly non-married women, to deal with economic insecurity since they face barriers to accessing formalised risk-coping strategies.

Commercial sex and transactional sex are attractive risk-coping strategies with the potential to make money quickly and earn up to 3 times more than alternative occupations [12]. The network of sexual partners developed through these activities may lead to other financial or in-kind gifts used to cope with economic shocks, defined as an unexpected, sudden, and significant change in a household's income or expenditure that, if left unaddressed, would have a significant impact on the household's consumption patterns [13,14]. Within these relationships,

women are incentivised to have unprotected sex because they receive a positive income premium for these risky sexual acts [15], ranging from 9% in Kenya to 66% more in India [16–18].

Recent evidence suggests that income variability is more important than income level for HIV transmission, with sudden economic shocks consistently leading to significant increases in risky sexual behaviours to meet basic needs [11,14,45]. In addition to income variability, poverty is linked to certain behaviours that can lead to making risky or poor decisions through cognitive biases [19]. It is hypothesised that scarcity and poverty may lead individuals to prioritise meeting immediate goals over longer-term goals, through influencing their risk and time preferences. In this context, it is possible that individuals experiencing poverty may take sexual risks in order to meet their immediate consumption needs [19].

Use of cash transfers is one type of structural intervention used to address the underlying causes of HIV risk by improving income levels. Cash transfers increase income levels and could discourage risky sexual behaviour, especially if condition-based incentives such as school attendance or remaining HIV–negative are included. However, the evidence on their effectiveness in reducing HIV is mixed. Only 3 out of 8 studies found cash transfers to be effective in protecting against HIV and sexually transmitted infections (STIs) and only when conditions were set [20].

One possible explanation for this is that cash transfers may have limited effectiveness in protecting households against economic shocks and poverty if their value is too low or if they are received outside a period of economic shock [14]. Conversely, some interventions based on financial incentives have shown some success, including lottery-based financial incentives to remain STI-free [21]. Such financial incentive design has been proven to be effective since it targets individuals who have a high preference for risk and hence who are more likely to engage in risky sexual behaviours [21]. In addition, interventions to improve women's self-efficacy in negotiating condoms has also been found to be effective [22]. Self-efficacy is an important aspect of HIV risk that can be increased through education, open conversations, and self-affirmation [23,24]. A literature review conducted recently found that structural interventions, such as social empowerment and economic strengthening, were significantly effective in reducing self-stigma that impedes HIV testing, HIV prevention, and HIV treatment [25].

Building on this evidence, we implemented the POWER study (***Pro*tecting *w*omen from *eco*nomic shocks to fight HIV in *A*frica**) to fill this gap in knowledge by protecting women engaging in transactional and commercial sex against one the most common economic shocks experienced by households in sub-Saharan Africa: out of pocket health spending [26]. POWER is a randomised controlled trial that aims to reduce the impact of economic shocks by providing health insurance without co-payment to eliminate health expenses at the point of care for vulnerable women most at risk of HIV in Cameroon. Out-of-pocket health expenditure is a major economic shock in Africa: it is high cost, frequent, concentrated among the poor, and the most common economic shock experienced by urban households [27]. In urban Cameroon, 21% of households spend more than 10% of their income on health out-of-pocket health spending every year, and only 2.7% of women and 8.2% of men are covered by health insurance [28]. In this paper, we evaluate the effectiveness of this intervention in reducing economic shocks on risky sexual behaviour and HIV and STI incidence among women engaged in commercial or transactional sex in Yaoundé, Cameroon.

## 2. Methods

The protocol paper with more details on the methods can be found in the supporting information [29]. This study is reported as per the Consolidated Standards of Reporting Trials (CONSORT) guideline for randomised controlled trials (S1 Checklist) presented in Fig 1.

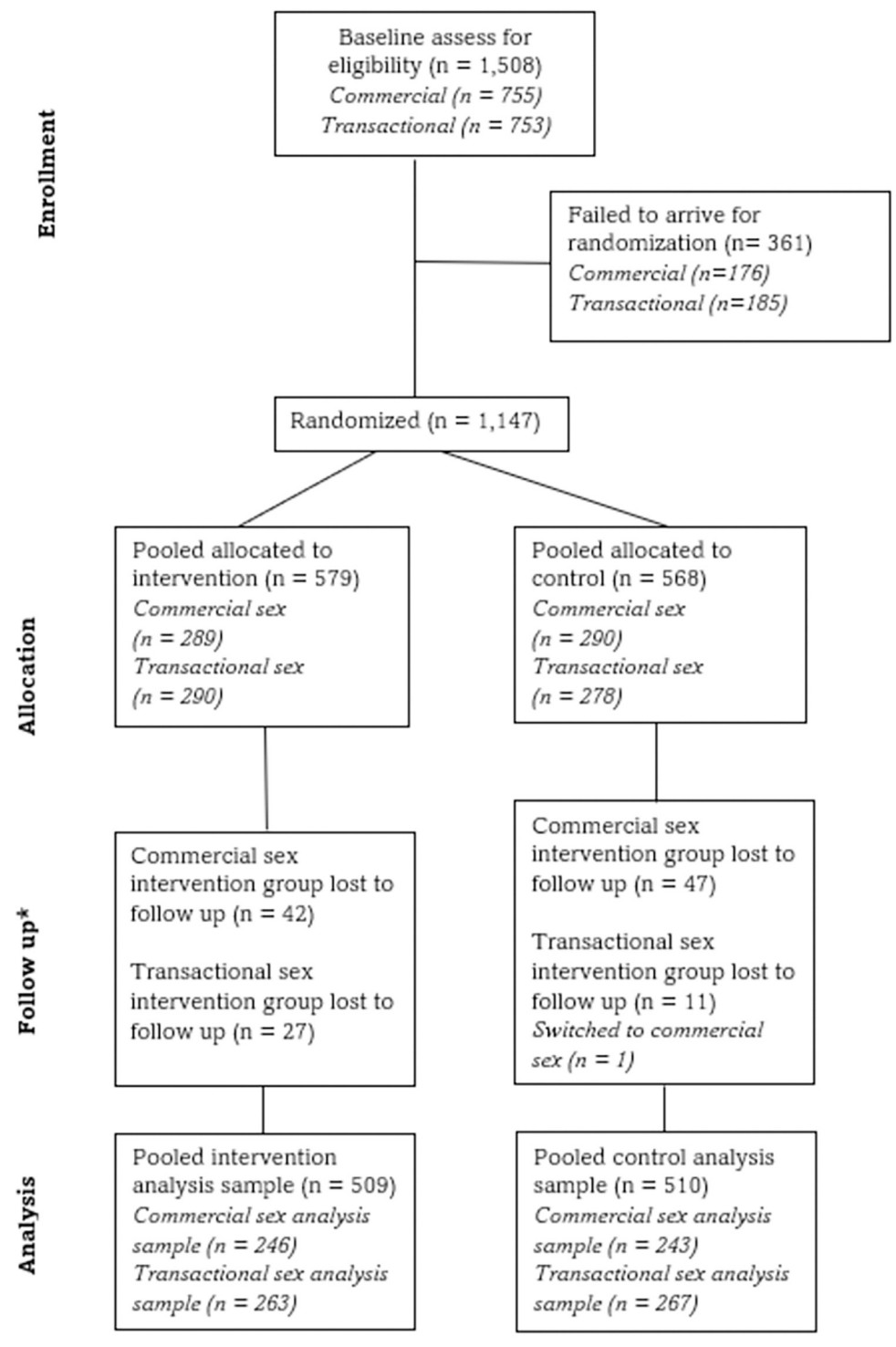

**Fig 1. CONSORT diagram.**

## 2.1. Recruitment and ethics

A stratified randomised controlled trial was conducted in Yaoundé, Cameroon, a country chosen due to having the highest HIV gender gap in the world. The sample was stratified 1:1 by the type of risky sexual activity: commercial or transactional sex. Two community-based organisations (CBOs) were identified, each working exclusively with one of the populations.

It should be noted that there is a blurred line between commercial and transactional sex [5]. The initial categorisation was based on peer and self-identification, which was then verified by members of the research team according to the types of relationships in which the participants were involved. The research team members responsible for recruitment were trained to categorise participants according to the following definitions. Transactional sex was defined as noncommercial, nonmarital sexual relationships motivated by the expectation that sex will be exchanged for material help, gifts, support, or other benefits. Transactional sex may often involve the exchange of money but is less systematic and explicit than commercial sex. Women who engage in transactional sex do not identify as female sex workers (women and girls engaged in commercial sex) [30] and typically have "sugar daddies," which we defined as a man who provides money, gifts, or other financial benefits to a younger woman in exchange for sex. Sugar daddies ("papy" in French) may be considered boyfriends by women, as transactional sex may take place in the context of romantic relationships. Conversely, commercial sex was defined as sexual relationships in which the sex act is directly remunerated, where the remuneration is usually always pre-negotiated and the sex acts do not take place in the context of romantic relationships but are conceived and perceived in terms of a "commodity exchange" in a professional environment. Women and girls engaged in commercial sex often identify themselves as such [31].

CBO "Renata" was chosen to recruit and follow up women who engage in transactional sex, while CBO "Horizon Femmes" was chosen for its expertise in working with women and girls engaged in commercial sex. CBO premises were used for data collection and the research team worked with CBO staff who followed up with study participants throughout the study. Snowball sampling was used to recruit participants for women and girls involved in both commercial and transactional sex, as these populations, particularly women and girls engaged in commercial sex, are often hidden due to the illegality of sex work in Cameroon [32]. Women were instructed to only recruit peers within their strata. In cases where there was uncertainty in identifying women and girls involved in commercial or transactional sex strata, the research teams categorised according to relationship type, using the definitions above. If a woman reported having at least 1 client in addition to sugar daddies, she was assigned to the commercial sex stratum.

The CBO identified 34 "seeds" for women and girls involved in transactional sex and 23 "seeds" for women and girls involved in commercial sex. Seeds had a large network within their community. Each of these seeds then recruited 3 participants who in turn were asked to recruit up to 3 other peers. This process was repeated over 4 steps of recruitment. An incentive of FCFA 500 (£0.64) was provided for each participant recruited who met the eligibility criteria. On average, 23 women and girls involved in transactional sex and 31 women and girls involved in commercial sex were recruited per seed, resulting in a total sample of 1,506 participants for the study.

To be eligible for the study, participants had to engage in transactional or commercial sex in Yaoundé, Cameroon, be female, aged 15 years or older, not be married, have at least 1 economic dependent living in Yaoundé, be HIV–negative, have a password-protected phone, and be able to respond to text messages. STI and HIV testing was conducted at baseline, with free treatment offered to those who tested positive, referral for antiretroviral therapy (ART)

initiation to those who tested positive for HIV, and referral to the CBO if the participant was severely depressed, a victim of physical violence, or had recently been raped.

Minors, defined as women recruited under the age of 21 in Cameroon, were included in the study. To protect their privacy and avoid negative consequences, the ethics committees allowed them to waive parental consent and obtain it from an adult representative of the CBOs. Further information can be found in the study protocol [29].

Ethical approval was granted by the UCL Ethics Committee (ref: 17341/00) and the National Ethics Committee in Cameroon (CNERSH, ref: 2020/12/1313). No adverse events occurred during the study.

## 2.2. Randomisation and blinding

Participant randomisation occurred between October and November 2021, i.e., up to 2 months after the baseline survey had taken place in order to minimise attrition (S1 Fig). Participatory randomisation was found to be preferable to computer-based randomisation during the formative research phase as it was perceived to be more transparent and fair to respondents [33]. Participants were told that they would all receive the intervention but that the intervention group would receive it the day of the randomisation, while the control group would receive the intervention once the study period had been concluded, i.e., 12 months later. Randomisation took place in a private room in the presence of 2 enumerators and a survey supervisor. A description of the randomisation procedure, eligibility checks, and informed consent for randomisation took place. Participants were then presented with a large, deep black bag and watched as the enumerator placed 2 coloured balls inside: an orange ball (for the intervention group) and a white ball (control group). To determine their group allocation, the participant drew a ball from the bag. Those who picked the orange ball (intervention group) were given a detailed description of the insurance product they would receive, asked to re-consent to participate in the study, and to name up to 6 economic dependents who would also be covered by the health insurance. Economic dependents were defined as any person who would rely on the study participant to pay for their out-of-pocket expenses; no family relationship was required for economic dependents. Those allocated to the control group were re-informed about all the services they could still benefit from if they decided to stay in the trial and were told again that they would receive the intervention at the end of the trial if they continued to participate. They were also offered counselling by the CBO to minimise frustration at not receiving the intervention. Although the counselling appeared to be popular in advance during the formative research, it was not widely taken up, with only 12% opting for it after allocation to the control group [33]. S1 Fig shows the project timeline with dates for baseline, randomisation and the 2 follow-up surveys.

Table 1 demonstrates successful randomisation. Any imbalances are type I errors and within the expectations of randomisation. By completing randomisation after baseline only for returning participants, we reduced the attrition rate but may have reduced the generalisability of our results.

## 2.3. Intervention

We designed an insurance product for the purpose of the research in collaboration with the national private insurance company Garantie Mutuelle des Cadres (GMC) Assurances. The insurance product covered the participant and up to 6 economic dependents nominated at randomisation. The health care was freely available to the selected participants and covered the costs of consultations, medicines, doctors' fees, and hospitalisation for non-chronic illnesses at a public hospital in Yaoundé, the Cité Verte Hospital. The hospital was chosen because of its

**Table 1. Summary characteristics at baseline.**

| | Commercial sex | | | | | | Transactional sex | | | | | |
| | Intervention | | | Control | | | Intervention | | | Control | | |
| | Obs | N | Mean / % | Obs | N | Mean / % | Obs | N | Mean / % | Obs | N | Mean / % |
|---|---|---|---|---|---|---|---|---|---|---|---|---|
| Age, years | 286 | - | 31.2 | 287 | - | 31.0 | 290 | - | 25.1 | 278 | - | 24.3 |
| Experience - months in sex work | 289 | - | 56.6 | 290 | - | 54.9 | 289 | - | 38.8 | 278 | - | 36.0 |
| **Nationality** | | | | | | | | | | | | |
| Cameroonian (%) | 289 | 288 | 100% | 290 | 288 | 99% | 290 | 290 | 100% | 278 | 277 | 100% |
| **Education** | | | | | | | | | | | | |
| Currently studying | 289 | 41 | 14% | 290 | 56 | 19% | 290 | 129 | 44% | 278 | 136 | 49% |
| **Highest achieved level of education** | | | | | | | | | | | | |
| Primary | 288 | 43 | 15% | 290 | 41 | 14% | 290 | 24 | 8% | 278 | 19 | 7% |
| Secondary primary cycle | 288 | 104 | 36% | 290 | 86 | 30% | 290 | 70 | 24% | 278 | 70 | 25% |
| Secondary second cycle | 288 | 82 | 28% | 290 | 91 | 31% | 290 | 112 | 39% | 278 | 91 | 33% |
| Superior | 288 | 55 | 19% | 290 | 63 | 22% | 290 | 80 | 28% | 278 | 93 | 33% |
| No education | 288 | 4 | 2% | 290 | 1 | 3% | 290 | 3 | 0% | 278 | 2 | 1% |
| Other | 288 | 0 | 0% | 290 | 8 | 0% | 290 | 1 | 1% | 278 | 3 | 1% |
| **Marital status** | | | | | | | | | | | | |
| Never married | 289 | 258 | 89% | 290 | 257 | 89% | 290 | 282 | 97% | 278 | 272 | 98% |
| Separated | 289 | 20 | 7% | 290 | 19 | 7% | 290 | 5 | 2% | 278 | 5 | 2% |
| Divorced | 289 | 3 | 1% | 290 | 7 | 2% | 290 | 0 | 0% | 278 | 0 | 0% |
| Widowed | 289 | 8 | 3% | 290 | 7 | 2% | 290 | 3 | 1% | 278 | 1 | 0% |
| **Economic variables** | | | | | | | | | | | | |
| Household size | 289 | - | 4.3 | 290 | - | 4.4 | 290 | - | 5.7 | 278 | - | 5.6 |
| Number of economic dependents | 289 | - | 3.1 | 290 | - | 3.1 | 290 | - | 2.9 | 278 | - | 3.0 |
| Number of child economic dependents | 289 | - | 1.8 | 290 | - | 1.9 | 290 | - | 1.6 | 278 | - | 1.9 |
| Have at least 1 child economic dependents | 289 | 235 | 81% | 290 | 254 | 88% | 290 | 239 | 82% | 278 | 241 | 87% |
| Number of biological children | 289 | - | 2.1 | 290 | - | 1.9 | 290 | - | 1.2 | 278 | - | 1.2 |
| Household expenditure per adult equivalent | 289 | - | 54,804 CFA | 290 | - | 47,426 CFA | 290 | - | 32,933 CFA | 278 | - | 38,776 CFA |
| **Sex work and behaviours** | | | | | | | | | | | | |
| Number of clients/sugar daddies in typical week | 268 | - | 8.5 | 265 | - | 8.2 | 290 | - | 2.8 | 278 | - | 2.6 |
| Number of sex acts in a typical week | 286 | - | 17.9 | 288 | - | 17.4 | 290 | - | 4.0 | 278 | - | 3.8 |
| Average earnings from sex work | 289 | - | 92,332 CFA | 290 | - | 93,748 CFA | 290 | - | 45,048 CFA | 278 | - | 42,986 CFA |
| Used PrEP before | 289 | 25 | 9% | 290 | 19 | 7% | 290 | 0 | 0% | 278 | 1 | 0% |
| Had an HIV test in the last 12 months | 288 | 267 | 93% | 290 | 268 | 92% | 290 | 167 | 58% | 277 | 157 | 57% |
| Used condom at last sex - direct question | 289 | 245 | 85% | 290 | 249 | 86% | 290 | 144 | 50% | 278 | 140 | 50% |
| Used condom at last sex - list experiment | 578 | - | 81% | 580 | - | 80% | 580 | - | 44% | 556 | - | 46% |

Obs, Observations; *N*, number of affirmative observations; PrEP, pre-exposure prophylaxis.

favourable location, fees, capacity, and reputation. The intervention did not cover COVID-19, chronic diseases, or maternal health care.

The insurance product was implemented as a prepaid scheme, meaning that the value of claims could not exceed the income from premiums. The hospital recorded and received reimbursement from the insurance company for each cost incurred by POWER participants allocated to the intervention group. Participants and their allocated economic dependents had a combined limit of CFAF 500,000 (£644.02 or $776.59 as of 18 August 2022) in health care costs over the 12 months. The insurance product started on 15 November 2021 and ended on

14 November 2022, with 2,428 people (579 participants allocated to the intervention group and their 1,849 economic dependents) eligible for free health care.

To prevent fraudulent use, insurance cards were generated and distributed to those eligible for the health insurance, which had to be presented along with an Identification (ID) document to receive free health care. ID cards were produced by our research team and included a photograph taken on site for participants without valid ID. To minimise stigma at the hospital, women were described as economically vulnerable without mentioning that they were involved in commercial or transactional sex. Training was provided to hospital staff to minimise stigma associated with other participant characteristics.

## 2.4. Outcomes

The primary outcome measure was HIV incidence, measured by rapid serological blood tests, with positive and indeterminate tests retested by enzyme-linked immunosorbent assay (ELISA), and STI incidence, including chlamydia, gonorrhoea, trichomoniasis, and syphilis. Syphilis was tested using 2 rapid tests (Venereal Disease Research Laboratory "VDRL" and Treponema pallidum haemagglutination "VDRL and TPHA" tests) to identify never-infected, inactive infections (past and treated or recent infections) and active infections (untreated, current infections). All active infections were treated at baseline, so we grouped non-active and never-infected together as syphilis negative. Gonorrhoea and chlamydia were tested using rapid tests, but trichomoniasis was tested in the laboratory as there was no rapid test available. As per protocol, all positive or inconclusive STI tests from the site were referred for confirmatory testing at the laboratory, and in all cases we prioritised the laboratory results in our analysis [29].

Outcomes lying on the path from the intervention to primary outcomes (HIV) were grouped into behavioural outcomes, including risky behaviours and intensity of commercial or transactional sex (how much sex women and girls engage in), with additional secondary outcomes of stress, loneliness, and violence experienced reported alongside [34,35]. This paper only examines outcomes directly related to health and risky behaviours. The secondary outcomes listed in the protocol are the focus of other papers and include poverty, mental and physical health outcomes, discrimination, self-efficacy, HIV knowledge, social networks, education, beliefs, attitudes, and preferences related to risky sexual behaviour.

We hypothesised that the intervention could affect HIV infection through 2 main channels (Fig 2). The first is that health insurance has been shown to increase health care utilisation [36,37] and may therefore lead to an increase in STIs treated, thereby reducing susceptibility to HIV given the existing epidemiological synergy between STIs and HIV [46]. However, ex-ante moral hazard may lead to an increase in risky behaviour since the intervention could reduce the costs associated with STI infection. The second channel is through the reduction in economic shocks resulting from the elimination of out of pocket medical payments that should prevent the need to engage in risky sexual behaviours [14,39]. Specifically, an increased likelihood of leaving the risky sex market (extensive margin outcome), a reduction in the number of sexual acts and sexual partners (intensive margin outcome), or a reduction in the likelihood of engaging in risky sex (unprotected and anal sex), which could lead to a reduction in the incidence of STIs and HIV.

It is worth noting that risky sexual behaviours such as unprotected sex are prone to sensitivity bias [47–49]. To minimise social desirability bias, we ensured the privacy of the interviews, recruited enumerators from the same community and population as the study participants, and collected data on unprotected sex using the double-list experiment method (see Appendix A in S1 Appendix for details).

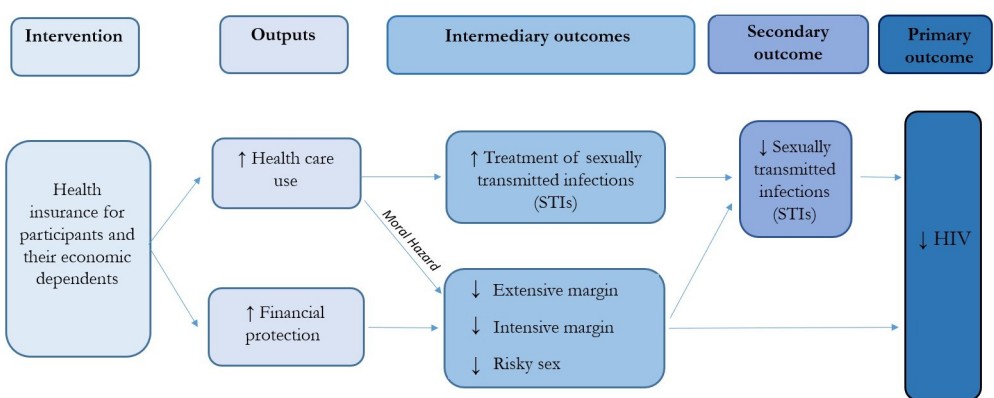

**Fig 2. Theory of change.** Note: A theory of change depicting the expected effects of the intervention.

## 2.5. Sample size and attrition

Power calculations were performed to determine the required sample size and minimum detectable effect. We assumed that 600 of the 750 participants recruited at baseline would be retained at 12 months post-intervention in both strata. With 80% power and a 5% significance level, and assuming that 20% of participants would acquire at least 1 STI during the study period, the sample size was sufficient to detect a 30% (6 percentage points) reduction in STI incidence in the intervention group. Analysing the subgroups of women engaged in commercial and transactional sex separately, the minimum detectable effect is a reduction in STI incidence of approximately 40% (8.3 percentage points). We initially assumed an HIV incidence of 4% over the study period, which meant that the sample size would allow us to detect a 65% reduction in HIV incidence (2.6 percentage points). These attrition assumptions were close to what we achieved in the trial. A total of 956 (493 transactional and 463 commercial) respondents completed the survey and biological tests at 6 months post-intervention and 806 (403 transactional and 403 commercial) at 12 months post-intervention. A total of 1,020 (530 transactional and 490 commercial) respondents completed the survey and biological tests at 6 months post-intervention or 12 months post-intervention, meaning that at least 1 post-intervention survey was available for 89% of the randomised sample.

Table 2 shows the response rates in each arm and strata of the study. The attrition rate for those who did not return for either 6 months post-intervention or 12 months post-intervention was very low, 7% for the women and girls involved in transactional sex and 16% for women and girls involved in commercial sex. We found a statistically significant difference in attrition between the intervention and control groups in women and girls involved in transactional sex: the control group was 5 percentage points more likely to return for follow-up than the intervention group. We found no difference in risky sexual behaviour at baseline between participants who attrited and those who did not in women and girls involved in transactional sex (Appendix B in S1 Appendix). In general, we found that those lose to follow up were younger and more likely to be in the commercial sex arm. However, we did not find a statistically significant difference between those lost to follow up in the intervention and control arms of the survey in both women and girls involved in commercial and transactional sex (Appendices C and D in S1 Appendix).

Our final sample of participants for analysis used information from the 12 months post-intervention survey (12 months post-intervention) or the 6 months post-intervention survey (6 months post-intervention) for those who did not complete the 12 months post-intervention

**Table 2. Response rates and attrition by arm.**

| | | | Pooled | | | | | |
|---|---|---|---|---|---|---|---|---|
| | Combined | | | Intervention | | | Control | |
| Baseline and randomisation | 1,146 | (100%) | | 568 | (100%) | | 579 | (100%) |
| 6 months post intervention | 956 | (83%) | | 478 | (84%) | | 478 | (83%) |
| 12 months post intervention | 806 | (70%) | | 394 | (69%) | | 411 | (71%) |
| 6 or 12 months post intervention | 1,020 | (89%) | | 510 | (90%) | | 510 | (88%) |
| | | | Transactional | | | | | |
| | Combined | | | Intervention | | | Control | |
| Baseline and randomisation | 568 | (100%) | | 290 | (100%) | | 278 | (100%) |
| 6 months post intervention | 493 | (87%) | | 246 | (85%) | | 247 | (89%) |
| 12 months post intervention | 403 | (71%) | | 193 | (67%) | | 210 | (76%) |
| 6 or 12 months post intervention | 530 | (93%) | | 263 | (91%) | | 267 | (96%) |
| | | | Commercial | | | | | |
| | Combined | | | Intervention | | | Control | |
| Baseline and randomisation | 579 | (100%) | | 289 | (100%) | | 290 | (100%) |
| 6 months post intervention | 463 | (80%) | | 232 | (80%) | | 232 | (80%) |
| 12 months post intervention | 402 | (69%) | | 201 | (70%) | | 201 | (69%) |
| 6 or 12 months post intervention | 489 | (84%) | | 246 | (85%) | | 243 | (84%) |

survey (row 4 of Table 2). This final analysis sample was the one we used throughout the paper, and all adjusted regressions took into account whether the information used was collected at 6 months post-intervention or 12 months post-intervention.

## 2.6. Statistical analysis

Authors, HC and AL led the analysis and were blinded to respondents' intervention allocation. All analyses were conducted in STATA version 17 and followed the analysis plan specified in the trial protocol [29] and as registered in the ISRCTN (trial 225165484). To assess the effect of health insurance on our primary and secondary outcomes, we conducted a modified intention-to-treat (ITT) analysis, where respondents were analysed based on their randomised trial group, but those with missing responses were not included. We estimated logistic regressions with odds ratios (OR) for our primary biological and all binary outcomes, reporting the estimated difference in proportions or the estimated marginal effect. For continuous outcomes, we estimated marginal effects using ordinary least squares (OLS). Table notes indicate which outcomes were estimated using which method. Adjusted models control for outcome level at baseline, logged adult equivalent total expenditure as a proxy for income, number of economic dependents, household size and whether the respondent had a child.

## 3. Results

We recruited 1,508 adolescent girls and women over 15 years old between June and August 2021 involved in transactional sex ($n = 753$) or commercial sex ($n = 755$) in Yaounde, Cameroon, using snowball sampling. Participants were randomly assigned (1:1) to receive free health insurance for themselves and their economic dependents for 12 months either at the beginning of the study (intervention; $n = 579$) from November 2021 or at the end of the study (control; $n = 568$) (Fig 1). A total of 938 medical visits took place over 12 months, including 411 participant patients (44%) and 527 economic dependent patients (56.1%). The main reasons for consultations included malaria (34%), STI treatment based on syndromic case management

(14%), cold and flu (14%), gastrointestinal illness (11%), skin problems (5%), lung infection (8%), pelvic pain (7%), and accident and injury (3%) (S2 Fig). For participants (excluding economic dependents), there were 123 consultations (30%) for STIs.

### 3.1. Effect of health insurance on HIV and other STIs

We found the intervention to be effective to prevent HIV. The intervention significantly reduced the odds of acquiring HIV for women and girls involved in transactional sex (OR = 0.109 (95% confidence interval (CI) [0.014, 0.870]); $p = 0.036$; adjusted odds ratio (AOR) = 0.113 (95% CI [0.014, 0.912]); $p = 0.041$) and in both strata combined (OR = 0.197, (95% CI [0.043, 0.905]); $p = 0.037$; AOR = 0.204 (95% CI [0.044, 0.940]); $p = 0.041$). Results are reported in Fig 3 and Appendix E in S1 Appendix (odd ratio) and Appendix F in S1 Appendix (marginal effects). There was no impact on HIV for women engaging in commercial sex, but as shown in Appendix G in S1 Appendix, the number of HIV infections was low over the period among this strata.

We found no effect of the intervention for other STIs (Appendices E and F in S1 Appendix). The absence of effect could be due to serious problems found with the STI tests because they were not effective in determining STI status. Every 15th chlamydia, syphilis, and HIV test

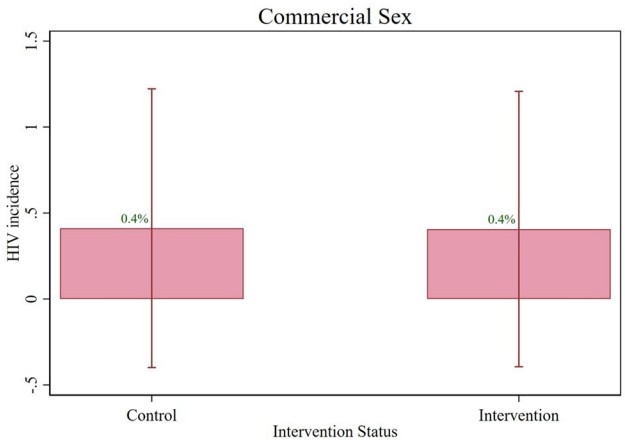

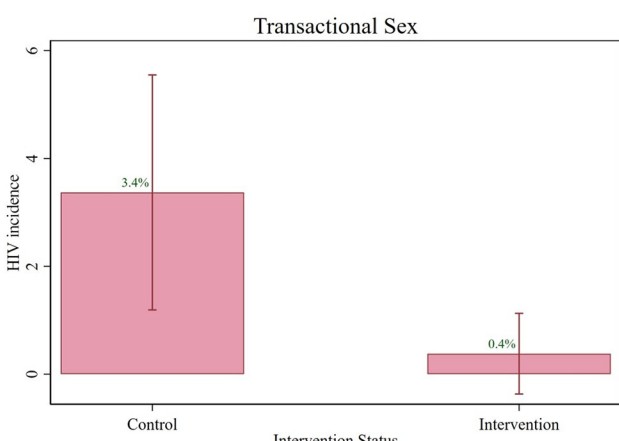

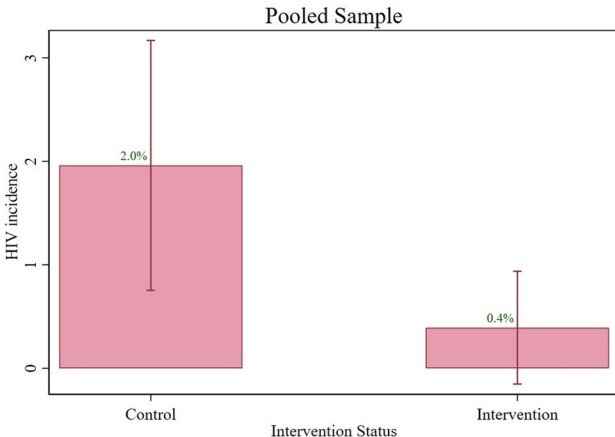

**Fig 3. Effect of free health insurance on HIV incidence.** Note: Women in commercial sex, $n = 489$; women and girls in transactional sex, $n = 530$; pooled sample, $n = 1,019$. Boxes indicate the HIV incidence measured at 6 or 12 months post-intervention and whiskers the 95% confidence interval of these means.

**Table 3. Result of rapid test vs. laboratory ELISA tests.**

|  | Rapid test | ELISA laboratory test |
|---|---|---|
| **HIV** |  |  |
| Positive | 5 | 5 |
| Negative | 57 | 58 |
| Inconclusive | 1 | 0 |
| **Chlamydia** |  |  |
| Positive | 1 | 57 |
| Negative | 202 | 149 |
| Inconclusive | 11 | 8 |
| **Syphilis** |  |  |
| Positive active infection | 2 | 0 |
| Negative | 49 | 63 |
| Positive - inactive infection | 0 | 11 |
| Inconclusive | 23 | 0 |

Note: only endline (12 months post-intervention) samples that had both a rapid and ELISA tests are included.

performed, as well as all positive and inconclusive tests, were verified in the laboratories using the same rapid tests. At 12 months post-intervention, as many samples as possible were retested in the ELISA laboratory using new types of assays. Table 3 shows large type II errors in the detection of chlamydia, imprecision in the detection of syphilis, but consistent results for HIV. There was evidence that the tests for chlamydia and syphilis lacked sensitivity and that there were problems with sample collection, transport, and storage for gonorrhoea and trichomoniasis tests in the field. Appendix H in S1 Appendix contains further details.

## 3.2. Effect on risky sexual behaviours

Table 4 shows the association between the intervention and risky sexual behaviour and other secondary outcomes. The intervention was associated with a significant increase in the likelihood of stopping transactional sex by 8.6 percentage points (16.9% versus 25.5%) (marginal effect (ME) = 0.086 (95% CI [0.015, 0.158]); $p = 0.018$). There was an increase in condom use at last sex of about 15 percentage points as measured by the list experiment method that was not statistically significant at the 5% level (ME = 0.152 (95% CI [−0.021, 0.324]); $p = 0.085$). We also observed a reduction in the number of sex acts by an average of 0.36 for women having transactional sex that was not statistically significant at the 5% level (ME = −0.358 (95% CI [−0.781, 0.065]); $p = 0.097$).

Among women and girls engaged in commercial sex, results were mixed. The intervention was not associated with a greater likelihood of leaving sex work or of engaging in protected sex. However, it was associated with very small increases in the likelihood of engaging in anal sex (ME = 0.014 (95% CI [0.002, 0.026]); $p = 0.024$) and in engaging in sex with riskier clients (ME = 0.037 (95% CI [−0.004, 0.078]); $p = 0.075$).

The study found that the intervention was not associated with the likelihood of experiencing physical violence nor with participants' levels of loneliness and stress. Both unadjusted and adjusted odd ratios are presented in Appendix I in S1 Appendix.

## 3.3. Analysis of the intervention channels and alternative explanations

A key question about this intervention is the pathway and mechanism by which it works. As shown in Fig 1, its impact on HIV may occur through two potential channels: (1) by reducing

**Table 4. Effect of health insurance on sexual behaviours behaviour outcomes.**

| Sexual Behaviour and secondary outcomes | Baseline | | | | | | Midline/endline | | | | | | | | | Difference | | |
| --- | --- | --- | --- | --- | --- | --- | --- | --- | --- | --- | --- | --- | --- | --- | --- | --- | --- | --- |
| | Intervention | | | Control | | | Intervention | | | Control | | | ME | p-value | 95% CI | ME | p-value | 95% CI |
| | Obs | N | %/mean | Obs | N | %/mean | Obs | N | %/mean | Obs | N | %/mean | | | | | | |
| **Commercial sex** | | | | | | | | | | | | | | | | | | |
| Number of regular clients in the last 7 days | 287 | - | 3.84 | 289 | - | 3.89 | 180 | - | 2.96 | 185 | - | 2.99 | -0.034 | | -0.585 , 0.518 | -0.036 | | -0.587 , 0.516 |
| Number of occasional clients in the last 7 days | 281 | - | 0.05 | 282 | - | 4.54 | 180 | - | 3.25 | 184 | - | 2.82 | 0.429 | | -0.306 , 1.165 | 0.382 | | -0.348 , 1.113 |
| Number of clients in the last 7 days (combined) | 279 | - | 8.85 | 281 | - | 8.46 | 180 | - | 6.21 | 184 | - | 5.74 | 0.129 | | -0.721 , 0.979 | 0.112 | | -0.711 , 0.936 |
| Number of sex acts in the last 7 days | 287 | - | 11.75 | 290 | - | 11.12 | 246 | - | 5.60 | 243 | - | 5.84 | -0.242 | | -1.352 , 0.868 | -0.364 | | -1.447 , 0.718 |
| Number of vaginal sex acts over previous 2 clients‡ | 578 | 562 | 97.2% | 580 | 567 | | 360 | 356 | 98.9% | 370 | 365 | 97.8% | 0.002 | | -0.014 , 0.018 | 0.002 | | -0.014 , 0.018 |
| Number of oral sex acts over previous 2 clients‡ | 574 | 69 | 12.0% | 580 | 93 | 16.0% | 359 | 34 | 9.5% | 370 | 52 | 16.0% | -0.046 | | -0.093 , 0.001 | -0.036 | | -0.082 , 0.01 |
| Number of anal sex acts over previous 2 clients‡ | 577 | 12 | 2.1% | 580 | 15 | 2.6% | 360 | 5 | 1.4% | 370 | 0 | 2.6% | 0.014 | | 0.002 , 0.026 | 0.013 | | 0.001 , 0.025 |
| Self-reported riskiness of clients (average /10)‡ | 578 | - | 0.38 | 580 | - | 0.38 | 360 | - | 0.59 | 380 | - | 0.55 | 0.039 | | 0.009 , 0.070 | 0.038 | | 0.007 , 0.068 |
| Used condom at last sex act - direct questioning† | 289 | 245 | 84.8% | 290 | 249 | 85.9% | 180 | 161 | 89.4% | 184 | 162 | 88.0% | 0.014 | 0.673 | -0.051 , 0.079 | 0.013 | 0.686 | -0.051 , 0.077 |
| Quit commercial sex work† | - | - | | - | - | | 218 | 49 | 22.5% | 221 | 48 | 21.7% | 0.008 | 0.848 | -0.070 , 0.085 | 0.008 | 0.824 | -0.067 , 0.084 |
| Suffered violence from sexual partner | 289 | 83 | 28.7% | 290 | 124 | 42.8% | 246 | 79 | 32.1% | 243 | 79 | 32.5% | -0.004 | 0.925 | -0.087 , 0.079 | 0.003 | 0.945 | -0.081 , 0.087 |
| Stress (PPS-4) | - | - | | - | - | | 246 | - | 7.98 | 243 | - | 7.98 | 0.004 | | -0.227 , 0.236 | - | | - , - |
| Loneliness (UCLA 3 point scale) | 289 | - | 4.97 | 290 | - | 5.11 | 246 | - | 5.67 | 243 | - | 5.60 | 0.066 | | -0.251 , 0.383 | 0.064 | 0.691 | -0.253 , 0.691 |
| **List experiment** | | | | | | | | | | | | | | | | | | |
| Used condom at last sex act - List experiment | - | - | | - | - | | - | - | | - | - | | -0.111 | | -0.348 , 0.125 | -0.109 | | -0.345 , 0.128 |
| **Transactional sex** | | | | | | | | | | | | | | | | | | |
| Number of regular papy in the last 7 days | 290 | - | 1.49 | 278 | - | 1.43 | 186 | - | 1.42 | 213 | - | 1.37 | 0.059 | | -0.190 , 0.307 | 0.080 | | -0.169 , 0.330 |
| Number of occasional papy in the last 7 days | 290 | - | 0.46 | 277 | - | 0.51 | 186 | - | 0.73 | 213 | - | 0.75 | -0.021 | | -0.191 , 0.150 | 0.003 | | -0.170 , 0.176 |
| Number of papy in the last 7 days | 290 | - | 1.95 | 278 | - | 1.94 | 263 | - | 1.52 | 267 | - | 1.69 | -0.164 | | -0.439 , 0.110 | -0.140 | | -0.414 , 0.133 |
| Number of sex acts in the last 7 days‡ | 290 | - | 2.64 | 278 | - | 2.74 | 263 | - | 2.13 | 267 | - | 2.48 | -0.358* | | -0.781 , 0.065 | -0.320 | | -0.742 , 0.102 |
| Number of vaginal sex acts‡ | 520 | 515 | 99.0% | 593 | 476 | 80.3% | 367 | 363 | 98.9% | 412 | 405 | 98.3% | 0.006 | | -0.011 , 0.024 | 0.007 | | -0.011 , 0.024 |
| Number of oral sex acts‡ | 520 | 68 | 13.1% | 493 | 87 | 17.6% | 366 | 47 | 12.8% | 414 | 64 | 15.5% | -0.023 | | -0.074 , 0.027 | -0.019 | | -0.068 , 0.030 |
| Number of anal sex acts‡ | 518 | 13 | 2.5% | 494 | 11 | 2.2% | 367 | 3 | 0.8% | 414 | 3 | 0.7% | 0.001 | | -0.012 , 0.014 | -0.001 | | -0.014 , 0.012 |
| Self-reported riskiness of clients (average /10)‡ | 520 | - | 0.47 | 495 | - | 0.48 | 358 | - | 0.35 | 400 | - | 0.34 | 0.014 | | -0.012 , 0.039 | 0.008 | | -0.018 , 0.034 |
| Used condom at last sex act - direct questioning† | 290 | 144 | 49.7% | 278 | 140 | 50.4% | 186 | 82 | 44.1% | 213 | 89 | 41.8% | 0.023 | | -0.074 , 0.120 | 0.016 | | -0.082 , 0.114 |
| Quit transactional sex work† | - | - | | - | - | | 247 | 63 | 25.5% | 249 | 42 | 16.9% | 0.086 | | 0.015 , 0.158 | 0.084 | | 0.013 , 0.158 |
| Suffered violence from sexual partner | 290 | 35 | 12.1% | 278 | 36 | 12.9% | 263 | 32 | 12.2% | 267 | 23 | 8.6% | 0.036 | | -0.017 , 0.088 | 0.036 | | -0.016 , 0.089 |
| Stress (PPS-4) | - | - | | - | - | | 263 | - | 8.05 | 267 | - | 8.04 | 0.012 | | -0.284 , 0.308 | - | | - , - |

*(Continued)*

**Table 4.** (Continued)

| Outcome | Obs | N | Mean | Obs | N | Mean | Obs | N | Mean | Obs | N | Mean | Unadjusted | Unadjusted 95% CI | Adjusted | Adjusted 95% CI |
|---|---|---|---|---|---|---|---|---|---|---|---|---|---|---|---|---|
| Loneliness (UCLA 3 point scale) | 290 | - | 5.82 | 287 | - | 5.60 | 263 | - | 5.35 | 267 | - | 5.25 | 0.099 | -0.179 , 0.038 | 0.101 | -0.181 , 0.384 |
| **List experiment** | | | | | | | | | | | | | | | | |
| Used condom at last sex act - list experiment | - | - | - | - | - | - | - | - | - | - | - | - | 0.152 | -0.021 , 0.324 | 0.150 | -0.025 , 0.325 |
| **Pooled (commercial + transactional)** | | | | | | | | | | | | | | | | |
| **Continuous outcomes - OLS** | | | | | | | | | | | | | | | | |
| Number of regular clients/papy in the last 7 days | 577 | - | 2.66 | 567 | - | 2.68 | 366 | - | 2.18 | 398 | - | 2.12 | 0.135 | -0.158 , 0.428 | 0.092 | -0.191 , 0.376 |
| Number of occasional clients/papy in the last 7 days | 571 | - | 2.7 | 559 | - | 2.55 | 366 | - | 2.33 | 397 | - | 2.04 | 0.233 | -0.077 , 0.544 | 0.167 | -0.132 , 0.466 |
| Number of papy's or clients/papy in the last 7 days | 579 | - | 5.32 | 568 | - | 5.18 | 509 | - | 3.24 | 510 | - | 3.24 | 0.030 | -0.398 , 0.457 | -0.010 | -0.418 , 0.397 |
| Number of sex acts in the last 7 days | 579 | - | 7.53 | 568 | - | 7.02 | 509 | - | 3.81 | 510 | - | 4.08 | -0.279 | -0.891 , 0.333 | -0.391 | -0.969 , 0.187 |
| Number of vaginal sex acts‡ | 1098 | 1077 | 98.1% | 1073 | 1043 | 97.2% | 727 | 719 | 98.9% | 782 | 770 | 98.5% | 0.004 | -0.007 , 0.016 | 0.005 | -0.007 , 0.017 |
| Number of oral sex acts‡ | 1094 | 137 | 12.5% | 1073 | 180 | 16.8% | 725 | 81 | 11.2% | 784 | 116 | 14.8% | -0.035 | -0.069 , -0.001 | -0.024 | -0.058 , 0.009 |
| Number of anal sex acts‡ | 1095 | 25 | 2.3% | 1074 | 26 | 2.4% | 727 | 8 | 1.1% | 784 | 3 | 0.4% | 0.007 | -0.001 , 0.016 | 0.007 | -0.002 , 0.016 |
| Self-reported riskiness of clients/papy (average/10)‡ | 1098 | - | 0.42 | 1075 | - | 0.43 | 718 | - | 0.47 | 770 | - | 0.44 | 0.031 | 0.008 , 0.054 | 0.027 | 0.005 , 0.049 |
| Used condom at last sex act - direct questioning† | 579 | 389 | 67.2% | 568 | 389 | 68.5% | 366 | 243 | 66.4% | 397 | 251 | 63.2% | 0.014 | -0.051 , 0.079 | 0.036 | -0.029 , 0.101 |
| Quit either commercial or transactional† | - | - | - | - | - | - | 465 | 112 | 24.1% | 470 | 90 | 19.1% | 0.008 | -0.070 , 0.085 | 0.043 | -0.008 , 0.095 |
| Suffered violence from sexual partner | 579 | 118 | 20.4% | 568 | 160 | 28.2% | 509 | 111 | 21.8% | 510 | 102 | 20.0% | 0.018 | -0.032 , 0.068 | 0.028 | -0.021 , 0.078 |
| Stress (PPS-4) | - | - | - | - | - | - | 509 | - | 8.02 | 510 | - | 8.01 | 0.008 | -0.182 , 0.197 | - | - |
| Loneliness (UCLA 3 point scale) | 579 | - | 5.4 | 568 | - | 5.35 | 509 | - | 5.51 | 510 | - | 5.42 | 0.085 | -0.125 , 0.296 | 0.085 | -0.127 , 0.296 |
| **List experiment** | | | | | | | | | | | | | | | | |
| Used condom at last sex act - list experiment | - | - | - | - | - | - | - | - | - | - | - | - | 0.034 | -0.113 , 0.180 | 0.041 | -0.106 , 0.187 |

Significance levels

* p < 0.1

** p < 0.05

*** p < 0.01.

Notes: Obs, = observations; N = affirmative observations; OLS = ordinary least squares; 95% CI = 95% confidence interval.

† Dependent binary variables (†) were estimated using logistic regression reporting marginal effects. All other continuous outcome variables were estimated using OLS reporting marginal effects. (log) indicate where the outcome is logged in the OLS model. Unadjusted = compares outcomes of follow-up data without adjustment. Adjusted = adjusts for baseline levels of the outcome, if the outcome was collected during midline, then baseline controls, namely log of adult equivalent expenditure, number of economic dependents, household size and if the respondent has a child. The list experiment was estimated using OLS using the double list experiment method [47], hence twice the observations as each respondent appears twice. PPS-4 is a measure of stress [34], and UCLA-3 of loneliness [35].

‡ = Data taken from the last 2 sex acts per sex worker.

biological susceptibility to HIV during unprotected sex through increased treatment of STIs; and (2) by reducing risky sexual behaviours through improved protection against economic shocks. To disentangle these 2 channels, we examine the impact of the intervention on STIs and economic shocks.

First, we can investigate the role of STIs by examining the role of self-reported STI symptoms in mediating the impact of the intervention on HIV in women and girls involved in transactional sex. Some STI symptoms that increase HIV susceptibility are noticeable by infected individuals, whereas HIV symptoms can remain hidden for years after infection. Of the 10 people who acquired HIV, one each from the intervention and control arms also reported STI symptoms, note we include potentially misreported urinary tract infections. This leaves 8 HIV infections, all in the control group, not associated with STIs, providing some evidence for the reduction in HIV through reduced risky behaviour channel rather than increased treatment of STIs caused by the intervention.

In addition, Appendix J in S1 Appendix shows the results when participants with STI symptoms were excluded, showing that the impact of the intervention still reduced HIV by approximately 3.4 percentage points when STI symptoms were excluded (ME = −0.034 (95% CI [−0.065, −0.003]); $p = 0.029$). The estimates of reported STI symptoms were low, so caution should be exercised in interpreting them. However, this provides further evidence that the intervention is not working through the STI treatment channel.

The final piece of evidence confirms that health expenditure and economic shocks were reduced by the intervention (Table 5). On average, 70% of insured participants or their economic dependents were sick at least once during the intervention period, and 57% of participants reported receiving free care for themselves or their economic dependents because of the intervention. In women and girls involved in transactional sex, the intervention reduced health expenditure on all measures except self-reported health expenditure shocks in the last 30 days. It also showed an increase in utilisation, but further statistical analysis of our primary outcomes conditional on utilisation cannot be performed because behaviours are likely to be influenced by the knowledge that future health care costs will be covered.

In Appendix K in S1 Appendix we also presented evidence that our results were not driven by ex-ante moral hazard, as the intervention did not increase the likelihood of reporting an illness, nor did it increase risky health behaviours.

## 4. Discussion

We found that health insurance provides against HIV acquisition for women engaged in transactional sex. We found strong evidence that the primary mechanism through which the intervention works is by protecting against health care spending related economic shocks that reduce risky sexual behaviours. Our findings were robust to removing HIV acquired through increased susceptibility from reported STI symptoms.

Our results showed the importance of including transactional sex in HIV transmission research, as we found 9 times more infections among women in transactional sex than among women in commercial sex in the control group. As there were fewer HIV infections among women and girls engaged in commercial sex than expected, the trial was underpowered to detect the effect of the intervention on HIV incidence among women and girls engaged in commercial sex. We did not find comparable reductions in self-reported risky sexual behaviour at the intensive and extensive margins among women and girls engaged in commercial sex, suggesting that the intervention did not reduce risky sexual behaviours, but this must be treated with caution. Although this paper cannot provide a complete explanation, we believe

**Table 5. Effect of health insurance on health utilisation and health spending outcomes.**

| Outcomes | | Unadjusted | | | | | | Adjusted | | | | |
|---|---|---|---|---|---|---|---|---|---|---|---|---|
| | *n* | ME | *p*-value | 95% CI | | | *n* | AME | *p*-value | 95% CI | | |
| **Commercial sex** | | | | | | | | | | | | |
| If sick since randomisation† | 489 | 0.068 | 0.129 | −0.020 | , | 0.155 | 489 | 0.068 | 0.128 | −0.020 | , | 0.156 |
| If respondent sought a health professional† | 228 | 0.039 | 0.455 | −0.063 | , | 0.140 | 228 | 0.031 | 0.549 | −0.070 | , | 0.132 |
| If at least 1 economic dependent reported sick† | 489 | −0.059 | 0.189 | −0.146 | , | 0.029 | 489 | −0.055 | 0.202 | −0.139 | , | 0.029 |
| Suffered health expense shock in the last 30 days† | 489 | −0.085 | 0.039 | −0.166 | , | −0.004 | 489 | −0.091 | 0.028 | −0.172 | , | −0.010 |
| Out of pocket payments in during the last illness (log CFAF) | 226 | −3.034 | <0.001 | −3.902 | , | −2.166 | 226 | −2.936 | <0.001 | −3.81 | , | −2.062 |
| Average spending on illnesses per month (log CFAF) | 489 | −0.787 | 0.006 | −1.346 | , | −0.228 | 489 | −0.758 | 0.008 | −1.317 | , | −0.198 |
| Out of pocket spending in the last 30 days (log CFAF) | 489 | −1.627 | <0.001 | −2.466 | , | −0.788 | 489 | −1.596 | <0.001 | −2.433 | , | −0.76 |
| **Transactional sex** | | | | | | | | | | | | |
| If sick since randomisation† | 530 | 0.031 | 0.758 | −0.054 | , | 0.115 | 530 | 0.033 | 0.448 | −0.052 | , | 0.118 |
| If respondent sought a health professional† | 290 | 0.126 | 0.008 | 0.032 | , | 0.219 | 290 | 0.135 | 0.004 | 0.042 | , | 0.228 |
| If at least 1 economic dependent reported sick† | 530 | −0.032 | 0.445 | −0.116 | , | 0.051 | 530 | −0.006 | 0.89 | −0.087 | , | 0.075 |
| Suffered health expense shock in the last 30 days† | 530 | −0.053 | 0.171 | −0.128 | , | 0.023 | 530 | −0.045 | 0.236 | −0.120 | , | 0.030 |
| Out of pocket payments in during the last illness (log CFAF) | 290 | −5.022 | <0.001 | −5.826 | , | −4.219 | 290 | −5.015 | <0.001 | −5.826 | , | −4.204 |
| Average spending on illnesses per month (log CFAF) | 530 | −0.579 | 0.049 | −1.155 | , | −0.003 | 530 | −0.619 | 0.037 | −1.199 | , | −0.038 |
| Out of pocket spending in the last 30 days (log CFAF) | 530 | −0.898 | 0.037 | −1.740 | , | −0.056 | 530 | −0.864 | 0.064 | −1.701 | , | −0.028 |
| **Pooled (commercial + transactional)** | | | | | | | | | | | | |
| If sick since randomisation† | 1,019 | 0.048 | 0.123 | −0.013 | , | 0.109 | 1,019 | 0.049 | 0.113 | −0.012 | , | 0.111 |
| If respondent sought a health professional† | 518 | 0.088 | 0.012 | 0.019 | , | 0.157 | 518 | 0.087 | 0.012 | 0.019 | , | 0.155 |
| If at least 1 economic dependent reported sick† | 1,019 | −0.046 | 0.139 | −0.107 | , | 0.015 | 1,019 | −0.031 | 0.297 | −0.090 | , | 0.028 |
| Suffered health expense shock in the last 30 days† | 1,019 | −0.068 | 0.016 | −0.123 | , | −0.013 | 1,019 | −0.065 | 0.022 | −0.120 | , | −0.009 |

(*Continued*)

**Table 5.** (Continued)

| Outcomes | Unadjusted | | | | | Adjusted | | | | |
|---|---|---|---|---|---|---|---|---|---|---|
| | n | ME | p-value | 95% CI | | n | AME | p-value | 95% CI | |
| Out of pocket payments in during the last illness (log CFAF) | 516 | −4.118 | <0.001 | −4.721 | , −3.514 | 516 | −4.09 | <0.001 | −4.695 | , −3.485 |
| Average spending on illnesses per month (log CFAF) | 1,019 | −0.678 | 0.001 | −1.080 | , −0.277 | 1,019 | −0.69 | 0.001 | −1.094 | , −0.286 |
| Out of pocket spending in the last 30 days (log CFAF) | 1,019 | −1.245 | <0.001 | −1.840 | , −0.651 | 1,019 | −1.209 | <0.001 | −1.800 | , −0.618 |

Notes: Obs = observations; *N* = affirmative observations; OLS = ordinary least squares; 95% CI = 95% confidence interval; ME = marginal effect; AME = adjusted marginal effect.

† Dependent binary variables (†) were estimated using logistic regression reporting marginal effects. All other continuous outcome variables were estimated using OLS reporting marginal effects. (log) indicate where the outcome is logged in the OLS model. Unadjusted = compares outcomes of follow-up data without adjustment. Adjusted = adjusts for baseline levels of the outcome, if the outcome was collected at 6 months post-intervention, then baseline controls, namely log of adult equivalent expenditure, number of economic dependents, household size, and if the respondent has a child. The recall period for if the respondent or dependent was sick and if they saw a health professional are "last 30 days" in baseline but "since randomisation" at 6 months post-intervention and 12 months post-intervention.

that the potentially heterogeneous impact could be explained by differences between the commercial and transactional sex contexts.

First, we believe that the extent to which women are entrenched in their activities differs. Women and girls engaged in commercial sex face greater barriers to exit than women in transactional sex. This is because they were more likely to depend on sex work for income, which is higher (93,000 CFAF versus 44,000 CFAF per month) and were less likely to have a second job as a result (31% versus 45%) or be enrolled in school (16% versus 46%) than women engaging in transactional sex. This means that the magnitude of the economic impact of the intervention was unlikely to be sufficient to force women and girls engaged in commercial sex out of the activity. In addition, given that health care costs account for a significant proportion of health care expenditure for women engaging in transactional sex (63% compared to 34% for women and girls engaged in commercial sex), if a sugar daddy were to offer to pay for medical bills, the removal of these costs for women could have been enough to end such activity.

Second, the intervention was less effective because HIV knowledge was higher among women and girls engaged in commercial sex, which leads to less scope for behaviour change. Women and girls engaged in commercial sex are considered a key population by UNAIDS [2] and have access to PrEP in Cameroon; among women and girls engaged in commercial sex in this study, 13% had ever taken PrEP at 12 months post-intervention, while none of the women engaging in transactional sex were using PrEP at 12 months post-intervention. In addition, over 90% of women and girls engaged in commercial sex had been tested for HIV in the last 12 months, compared with only 50% of women in transactional sex (Table 1). Third, women in transactional sex engaged in riskier sexual behaviour, albeit with fewer partners (Table 1) and may therefore have been more responsive to the intervention. In a companion paper, we show that trust and implied future support (compared to immediate payment for sex workers) are part of the payment system for transactional sex and may explain the higher levels of risky behaviour [38].

Third, stigma from health workers towards women and girls engaged in commercial sex is a main barrier to health care use and leads to self-stigma [42]. This may explain why there was no increase in the likelihood of visiting a professional among women and girls engaged in commercial sex compared to women who engage in transactional sex. This issue may limit the effect of hospital-based interventions for women and girls engaged in commercial sex.

This is the first study to our knowledge to highlight the critical heterogeneity of women who engage in transactional sex and female sex workers in the study of HIV prevention. Previous literature has often conflated commercial and transactional sex [5]. The more recent debate focuses on the similarities in their structural drivers [40,41] and questions the usefulness of having distinct categories [43]. However, this study quantitatively highlights the differential risks taken by women engaging in commercial and transactional sex, meaning HIV prevention needs to be aware of the nuanced differences. The literature should consider some degree of categorisation to understand the differential impacts of HIV prevention interventions among these separate groups.

Even among women and girls involved in transactional sex, HIV incidence among the control group over the 12 months was lower than expected at 2%, half of the official 4% statistics estimated in Cameroon. As a robustness check, we used bootstrapping techniques over 1,000 resamples and found similar results, Appendix E in S1 Appendix. The suspected issues with chlamydia and syphilis are important and worrying since the tests used in this study are the Cameroon Health Authority's recommended tests. Should they be inaccurate, it could indicate significant country-level type II error with major implications for the national HIV prevention strategy. Secondary outcomes were primarily self-reported, with the exception of condom use by the list experiment, which was indirectly elicited using a list experiment. It is important to note that self-reported outcomes may be subject to measurement error and should therefore be interpreted with caution.

The participants in our study may be subject to bias due to the limitations of the snowball methodology. Specifically, the seeds used by the CBO who initiated our recruitment chains may have led to the exclusion of participants with certain characteristics. As a result, the generalizability of our findings to the wider population of women engaged in transactional and commercial sex may be limited. Note that this recruitment strategy was the only option available, given that both strata were hidden populations.

A final potential limitation of the study was differential attrition between the intervention and control groups in women and girls involved in transactional sex. However, assuming that all participants in the intervention arm of women and girls involved in transactional sex were treated for HIV and assuming a conservative HIV incidence equal to the one in the control arm, it would result in less than 1 additional HIV infection in the intervention arm. Running the unadjusted logistic regression by adding an additional HIV infection in the intervention arm yields only a small reduction in estimated impact that remains statistically significant at 5%.

We contribute to the growing evidence that women engaging in transactional sex should be considered as a key population in the fight against HIV. We show how a structural intervention is more protective against HIV for women in transactional sex than women and girls engaged in commercial sex, yet currently they receive far fewer HIV prevention services and are ineligible to PrEP in many places [2]. Our study highlights the effectiveness of targeting studies at specific causes of economic vulnerability (i.e., economic shocks) where cash transfers have mixed impacts on HIV and risky behaviours. It is clear structural interventions alongside psychosocial interventions targeting stigma, self-efficacy and other causes of risky behaviours that could improve testing, identification and treatment of HIV have the potential to tackle the ongoing HIV epidemic in sub-Saharan Africa.

We performed a back-of-the-envelope calculation for the cost-effectiveness. The total cost of health care claimed by the intervention was £42,003, which averted 9 HIV infections in women and girls involved in transactional sex. This gives a cost-effectiveness ratio per HIV infection averted of £4,667 for women engaged in transactional sex. This means that the intervention is likely to be highly cost-effective when scaled up, and this figure does not consider the positive externalities associated with preventing HIV in the wider community.

Further research is necessary to gain a better understanding of the heterogeneous impacts between the 2 study populations. Additionally, a thorough estimation of the cost-effectiveness of the intervention to prevent HIV at the population level is important, taking into account the impact of the infections averted on other sexual partners. Furthermore, research is needed to determine the applicability of similar interventions for other common causes of economic shocks. For example, women in rural areas who are at risk of contracting HIV are often highly exposed to economic shocks caused by droughts which translate in risky sexual behaviours [44]. Therefore, drought insurance could be a suitable tool to prevent HIV in this context. Additionally, future research should focus on preventing HIV among women and girls engaged in commercial sex. For this group, combining this intervention with an intervention that enables women and girls engaged in commercial sex to substitute their earnings from sex work (e.g., cash transfer) might be more effective.

To conclude, our study provides, to the best of our knowledge, the first evidence of an intervention directly targeting health related economic shocks as a strategy to prevent HIV among women. Our study highlights the importance of including women engaging in transactional sex in Africa as a key population to receive additional support and protection in the fight against HIV.

## Supporting information

**S1 Checklist. CONSORT Checklist.**
(DOCX)

**S1 Fig. Project timeline.**
(TIF)

**S2 Fig. Reasons for consultation.**
(TIF)

**S1 Appendix. Appendix.**
(DOCX)

**S1 Data. Data set.**
(XLSX)

**S1 Study Protocol. Study Protocol.**
(DOCX)

## Author Contributions

**Conceptualization:** Aurélia Lépine, Sandie Szawlowski, Illiasou Mfochive.

**Data curation:** Aurélia Lépine, Sandie Szawlowski, Emile Nitcheu, Henry Cust.

**Formal analysis:** Aurélia Lépine, Henry Cust.

**Funding acquisition:** Aurélia Lépine.

**Investigation:** Aurélia Lépine, Sandie Szawlowski, Emile Nitcheu, Henry Cust, Illiasou Mfochive.

**Methodology:** Aurélia Lépine, Illiasou Mfochive.

**Project administration:** Aurélia Lépine, Sandie Szawlowski, Emile Nitcheu, Eric Defo Tamgno, Julienne Noo, Fanny Procureur, Serge Billong, Ubald Tamoufe.

**Supervision:** Aurélia Lépine, Emile Nitcheu, Eric Defo Tamgno, Julienne Noo, Serge Billong, Ubald Tamoufe.

**Writing – original draft:** Aurélia Lépine, Henry Cust.

**Writing – review & editing:** Aurélia Lépine, Sandie Szawlowski, Henry Cust, Illiasou Mfochive, Serge Billong, Ubald Tamoufe.

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
