## [Editor Report · Decision Letter 0]

23 Jan 2024

Dear Dr Lépine, 

Thank you for submitting your manuscript entitled "Protecting women from economic shocks to prevent HIV in Africa: Evidence from the POWER randomised controlled trial in Cameroon" for consideration by PLOS Medicine.

Your manuscript has now been evaluated by the PLOS Medicine editorial staff, and I am writing to let you know that we would like to send your submission out for external peer review.

Please re-submit your manuscript within two working days, i.e. by Jan 25 2024 11:59PM.

Feel free to email me directly at hvanepps@plos.org if you have any queries relating to your submission.

Kind regards,

Heather

Heather Van Epps, PhD

Executive Editor

PLOS Medicine

---

## [Decision Letter · Decision Letter 1]

28 Feb 2024

Dear Dr. Lépine,

Thank you very much for submitting your manuscript "Protecting women from economic shocks to prevent HIV in Africa: Evidence from the POWER randomised controlled trial in Cameroon" (PMEDICINE-D-24-00181R1) for consideration at PLOS Medicine. 

Your paper was evaluated by an associate editor and discussed among all the editors here. It was also discussed with an academic editor with relevant expertise, and sent to independent reviewers, including a statistical reviewer. The reviews are appended at the bottom of this email and any accompanying reviewer attachments can be seen via the link below:

[LINK]

In light of these reviews, I am afraid that we will not be able to accept the manuscript for publication in the journal in its current form, but we would like to consider a revised version that addresses the reviewers' and editors' comments. Obviously we cannot make any decision about publication until we have seen the revised manuscript and your response, and we plan to seek re-review by one or more of the reviewers. 

Please use the following link to submit the revised manuscript: https://www.editorialmanager.com/pmedicine/

We expect to receive your revised manuscript by Mar 20 2024. However, if this deadline is not feasible, please contact me by email, and we can discuss a suitable alternative.

Don't hesitate to contact me directly with any questions (aschaefer@plos.org). If you reply directly to this message, please be sure to 'Reply All' so your message comes directly to my inbox.

We look forward to receiving your revised manuscript. 

Sincerely,

Alexandra Schaefer, PhD

PLOS Medicine

plosmedicine.org

***Please note: not all will apply to your paper, but please check each item carefully

ACADEMIC EDITOR COMMENTS

The central reasons for needing a revision are to unpack the rationale for the study, put this in context of what is known, clarify that due to the design the findings may not be generalizable, but there is enough to work to understand the mechanism better. Alternative interventions could be considered in future trials such as guaranteed basic income. The term “shock” in this context was new to me so could be explained and the abstract edited to include a definition.

GENERAL COMMENTS

1) Please include page numbers and line numbers in the manuscript file. Use continuous line numbers (do not restart the numbering on each page). 

2) Please cite the reference numbers in square brackets. Citations should be preceding punctuation.

FINANCIAL DISCLOSURE 

The funding statement should include: specific grant numbers, initials of authors who received each award, URLs to sponsors’ websites. Also, please state whether any sponsors or funders (other than the named authors) played any role in study design, data collection and analysis, the decision to publish, or preparation of the manuscript. If they had no role in the research, include this sentence: “The funders had no role in study design, data collection and analysis, decision to publish, or preparation of the manuscript.”

COMPETING INTEREST

All authors must declare their relevant competing interests per the PLOS policy, which can be seen here:

https://journals.plos.org/plosmedicine/s/competing-interests

For authors with ties to industry, please indicate whether any of the interests has a financial stake in the results of the current study.

TITLE

Please revise your title according to PLOS Medicine's style. Your title must be nondeclarative and not a question. It should begin with main concept if possible. "Effect of" should be used only if causality can be inferred, i.e., for an RCT. Please place the study design ("A randomized controlled trial," "A retrospective study," "A modelling study," etc.) in the subtitle (ie, after a colon).

ABSTRACT

1) Please report your abstract according to CONSORT for abstracts, following the PLOS Medicine abstract structure (Background, Methods and Findings, Conclusions). https://www.equator-network.org/reporting-guidelines/consort-abstracts/

2) Please note that trial abstracts should only report secondary outcomes if all secondary outcomes of the trial are included. For trials that have many secondary outcomes, the abstract should be limited to reporting the primary outcome.

3) PLOS Medicine requests that main results are quantified with 95% CIs as well as p values. When reporting p values please report as p<0.001 and where higher as the exact p value p=0.002, for example. For the purposes of transparent data reporting, if not including the aforementioned please clearly state the reasons why not. When a p value is given, please specify the statistical test used to determine it. 

4) Throughout, suggest reporting statistical information as follows to improve clarity for the reader “22% (95% CI [13%,28%]; p</=)”. Please be sure to define all numerical values at first use. Please amend throughout the abstract and main manuscript (including tables). Please note the use of commas to separate upper and lower bounds, as opposed to hyphens as these can be confused with reporting of negative values.

5) Please ensure that all numbers presented in the abstract are present and identical to numbers presented in the main manuscript text.

6) Please include the study design, population and setting, number of participants, years during which the study took place (enrollment and follow up), length of follow up, and main outcome measures.

7) Please specify who was blinded to the intervention and control, define the intervention and control states, provide the number in each group, state that analysis was intention to treat and provide the number of participants lost to follow up in each group.

8) Please include the important dependent variables that are adjusted for in the analyses.

9) Please define all abbreviations including those for statistical reporting at first use.

10) Abstract Background: Provide the context of why the study is important. The final sentence should clearly state the study question.

11) In the last sentence of the Abstract Methods and Findings section, please describe the main limitation(s) of the study's methodology.

12) Abstract Conclusions: Please address the study implications without overreaching what can be concluded from the data and interpret the study based on the results presented in the abstract, emphasizing what is new without overstating your conclusions.

13) Please include the clinical trial registry number in the abstract.

AUTHOR SUMMARY

At this stage, we ask that you include a short, non-technical Author Summary of your research to make findings accessible to a wide audience that includes both scientists and non-scientists. The authors summary should consist of 2-3 succinct bullet points under each of the following headings:

• Why Was This Study Done? Authors should reflect on what was known about the topic before the research was published and why the research was needed.

• What Did the Researchers Do and Find? Authors should briefly describe the study design that was used and the study’s major findings. Do include the headline numbers from the study, such as the sample size and key findings.

• What Do These Findings Mean? Authors should reflect on the new knowledge generated by the research and the implications for practice, research, policy, or public health. Authors should also consider how the interpretation of the study’s findings may be affected by the study limitations. In the final bullet point of ‘What Do These Findings Mean?’, please describe the main limitations of the study in non-technical language.

Author Summary should immediately follow the Abstract in your revised manuscript. This text is subject to editorial change and should be distinct from the scientific abstract. Please see our author guidelines for more information: https://journals.plos.org/plosmedicine/s/revising-your-manuscript#loc-author-summary

METHODS AND RESULTS

1) PLOS Medicine requests that main results are quantified with 95% CIs as well as p values. We suggest reporting statistical information as detailed above – see under ABSTRACT

2) Please present numerators and denominators for percentages (at least in the Tables [not necessarily each time they're mentioned]).

3) Please complete the CONSORT checklist and ensure that all components of CONSORT are present in the manuscript, including [how randomization was performed, allocation concealment, blinding of intervention, definition of lost to follow-up, power statement]. When completing the checklist, please use section and paragraph numbers, rather than page numbers.

4) Please include the study protocol document and analysis plan, with any amendments, as Supporting Information to be published with the manuscript if accepted.

5) Please provide a flowchart (CONSORT flowchart may be used as a model) and indicate the number of individuals in each group analyzed in the ITT analysis.

6) The trial protocol lists “STI acquisition measured using % test positive for syphilis, % test positive for chlamydia, % test positive for gonorrhoea” as one of the two primary outcomes but it is listed as a secondary outcome in the current manuscript. Please clarify and explain this discrepancy.

7) The trial protocol lists several secondary outcomes of, such as violence, stress as measured by the Perceived Stress Scale 4 (PSS-4), or loneliness, which are not reported here. You state that the current manuscript only examines the outcomes relating directly to health and risky behavior outcomes. Please list all secondary outcomes in the Methods section and please indicate those that will be reported elsewhere. 

8) Please present the safety data for the study including numbers of specific events and whether or not adverse events are thought to be related to the intervention.

DISCUSSION

Please present and organize the Discussion as follows: a short, clear summary of the article's findings; what the study adds to existing research and where and why the results may differ from previous research; strengths and limitations of the study; implications and next steps for research, clinical practice, and/or public policy; one-paragraph conclusion.

FIGURES

For all Figures, please ensure that you have complied with our figures requirements http://journals.plos.org/plosmedicine/s/figures.

1) Please provide titles and legends for all figures (including those in Supporting Information files).

2) Please consider avoiding the use of red and green in order to make your figure more accessible to those with color blindness.

3) Please in the figure legend/description, define abbreviations used in each figure (including those in Supporting Information files).

4) Please define the meaning of all dots, lines and bars in the captions/footnotes.

TABLES

1) Please provide titles and legends for all tables (including those in Supporting Information files).

2) Please define all abbreviations used in the table below each table (including those in Supporting Information files). 

SUPPLEMENTARY MATERIAL

1) For supplementary figures and tables, please see the general comments under TABLES and FIGURES (color, abbreviations, titles, descriptions, etc.) and amend accordingly.

2) We suggest reporting statistical information as detailed above – see under ABSTRACT. Please be sure to define all numerical values.

3) As for the main manuscript, please indicate whether analyses are adjusted to help facilitate transparent data reporting please also detail the factors adjusted for and present the unadjusted analyses for comparison. If not, please clearly state the reasons why not.

4) Please revise the references in the supplementary material according to comments detailed below – see under REFERENCES.

5) Please cite your Supporting Information as outlined here: https://journals.plos.org/plosmedicine/s/supporting-information

REFERENCES

1) PLOS uses the numbered citation (citation-sequence) method and first six authors, et al.

2) Please ensure that journal name abbreviations match those found in the National Center for Biotechnology Information (NCBI) databases (http://www.ncbi.nlm.nih.gov/nlmcatalog/journals), and are appropriately formatted and capitalised.

3) Where website addresses are cited, please specify the date of access (e.g. [accessed: 12/06/2023]).

4) Please also see https://journals.plos.org/plosmedicine/s/submission-guidelines#loc-references for further details on reference formatting. 

5) RE reference [24] listed as “Working paper”, papers cannot be listed in the reference list until they have been accepted for publication or are publicly available on a preprint archive. 

Comments from the reviewers:

Reviewer #1: The paper entitled "Protecting Women from economic shocks to prevent HIV in Africa: Evidence from the POWER randomized controlled trial in Cameroon" deals with a very important topic in Sub-Saharan Africa. The topic is not novel; however, this could still be of interest since this is a new study population (women in Cameroon) and adds to the limited literature on this topic in this population. Furthermore, and most importantly, it is a study that presents the results of an intervention. However, the article needs some improvements:

- The introduction should better explain how the economic situation influences other psychosocial variables related to sexual behaviour. It is essential to explain the path that poverty takes before it leads to unprotective behaviour. Does poverty directly influence condom use, for example? Or does it influence the socio-cognitive variables that activate preventive behaviour? In the specific case of women, what role does poverty play in self-efficacy (an essential variable for negotiating safe sex with partners)? Read the following articles on HIV prevention in women (but not only, do an active search for articles of this nature):

1) Gender, HIV knowledge and prevention attitudes among adolescents living with HIV participating in an economic empowerment intervention in Uganda." AIDS care vol. 33,7 (2021): 888-896. doi:10.1080/09540121.2020.1844860

2) Socio-demographic, marital and psychosocial predictors of safe sex behaviour among Mozambican women at risk for HIV/AIDS. African Journal of AIDS Research, 17(4), 323-331. https://doi.org/10.2989/16085906.2018.1536672.

3) Socio-demographic, marital, and psychosocial factors associated with condom use negotiation self-efficacy among Mozambican women at risk for HIV infection. International Journal of Behavioral Medicine, 24(6), 846-855. https://doi.org/10.1007/s12529-017-9681-0.

- The methodology and results are presented in a very clear and well-articulated manner.

- In the discussion (and in the introduction) it is important to present the results of the effectiveness of other non-medical interventions (more social and psychosocial) in African women, particularly in sub-Saharan Africa. Are social and psychosocial interventions successful? If so, to what extent? What are its strengths and weaknesses? Here too, in the debate, more culturally diverse studies need to be presented. The following articles can help improve these two points:

1) Effectiveness of educational intervention among seropositive women about knowledge about HIV sexual transmission." Revista brasileira de enfermagem vol. 76,3 e20220371. 21 Aug. 2023, doi:10.1590/0034-7167-2022-0371

2) The Effect of Theory-Based HIV/AIDS Educational Program on Preventive Behaviors Among Female Adolescents in Tehran: A Randomized Controlled Trial." Journal of reproduction & infertility vol. 21,3 (2020): 194-206.

3) Testing the effectiveness of two psychosocial interventions - ACCENT and Didactic - to prevent HIV/AIDS behavioral risk factors in Mozambican women: a randomized controlled study. AIDS Care. https://doi.org/10.1080/13548506.2023.2229236.

4) Testing the effectiveness of two psychosocial interventions on socio-cognitive risk factors for HIV/AIDS in Mozambican Women: A randomized controlled trial. AIDS Education and Prevention, 33(3), 169-186. https://doi.org/10.1521/aeap.2021.33.3.169

5) Evaluating the effectiveness of incentives to improve HIV prevention outcomes for young females in Eswatini: Sitakhela Likusasa impact evaluation protocol and baseline results. BMC Public Health 20, 1591 (2020). https://doi.org/10.1186/s12889-020-09680-8

- After revising the manuscript, please make sure the abstract accurately summarizes it. 

- Thank you for the opportunity to read and review this article.

Reviewer #2: Statistical review

This paper reports a randomised trial investigating whether provision of health insurance reduces HIV incidence in individuals engaging in transactional and commercial sex.

The trial provides interesting results. I had some comments on the statistical methods and reporting:

1. Abstract: briefly mention what happened with the control group, especially that it involved receiving the intervention 12 months later.

2. Abstract: I would explicitly mention the primary outcome of the study.

3. Abstract (and results): for the primary analysis results, please add 95% CI for the odds ratio and put a more precise p-value following PLOS medicine guidance. I wasn't clear why the results for the transactional sex population were provided rather than the pooled results.

4. Abstract "We showed that the intervention allowed women to leave transactional sex." - perhaps add the proportions in each group that left?

5. Abstract: I would report secondary outcomes like proportion using condoms in the same way as the primary outcome (and add n in each group for this as it's not a randomised comparison given the difference in those leaving transactional sex.

6. Abstract "We showed that the intervention effectiveness operates through the reduction in health shocks since the increase in healthcare use following the intervention was low." - this sentence comes a bit out of the blue as there was no mention of use of healthcare or 'health shocks' in the results.

7. Abstract "We estimated that in our trial, the cost for each HIV infection averted is £4,667 among the cohort of women engaging in transactional sex." - this might be better in the results section, and ideally would have some uncertainty interval provided. (Similar comment applies to this in the discussion of the paper, although it is clearer there that it is an informal calculation - perhaps removing this statement from the abstract but keeping it in the discussion would be appropriate).

8. Table 1: given the groups were randomly allocated there's no need to test for imbalance between them as significant results must be type I errors. Thus, I'd remove the p-values (note that only 1 p-value out of >20 was significant, consistent with this being by random chance).

9. Table 2 - it would be useful to see whether the rates differed by randomised group. If so this may lead to more prospect of bias due to attrition (although the attrition rate is not that high, so I wouldn't be too worried about this).

10. Statistical analysis "we conducted an intention-to-treat (ITT) analysis where respondents were analysed based on their randomised study group" - I would probably refer to this as modified ITT or ITT with complete cases to account for the fact that individuals with missing response were not included.

11. Statistical analysis "We report the marginal effect for all other binary outcomes" - I wasn't quite sure what this means in comparison to the primary analysis results - can it be clarified? I presume it means the estimated difference in proportions rather than the OR. The methods section implies binary secondary outcomes would be analysed with a method for continuous outcomes but the footnote for table 4 says something different. I would recommend clarifying this in the methods section.

12. Table 4: the * and ** are not defined from what I can see - presumably p<0.05 and p<0.01; I would generally recommend the exact p-values are provided.

13. Page 17: I didn't follow the analysis used for results in Table A4 - if the authors are adjusting for post-baseline terms in a linear regression then I do not think this is the correct way to investigate whether one variable mediates another.

James Wason

Reviewer #3: 1) The eligibility criteria considered engagement in either transactional or commercial sex (see pg. 5), which seems to influence the results - showing positive outcomes in one category and none in the other. The reader may wonder how this study defined/differentiated the two in an African context (Yaunde).

2) Following the above concern, it is important to state - where such definitions come from to avoid imposition and epistemic injustices that have historically characterised global health practice.

3) It is interesting that the intervention allowed women to leave transactional sex and an increase in condom use and reduction in sex acts for those who remained in TS (pg. 15). However, these results are based on self-reports. Thus, they cannot form a solid basis for policy recommendations. Adding caution as part of the study's limitation can help clear some of these concerns. 

Reviewer #4: This is an interesting and important paper that explores the impact of protection from economic shocks on HIV prevention amongst commercial sex workers and women engaged in transactional sex in Cameroon. The paper contributes to the growing literature on women at the edge of sex work but who do not identify as selling sex and emphasises the importance of their inclusion within HIV prevention strategies including PrEP. The paper also suggests that protection from economic health shocks is likely to be of benefit in preventing HIV amongst women involved in transactional sex, but less protective for female sex workers, a finding that is not unsurprising but this is the first time a randomised trial has shown demonstrated this. I have some minor comments in relation to the methods and some suggestions where further clarification might be helpful to the reader.

Whilst I recognise the Respondent Driven Sampling approach has been detailed elsewhere in a UCL protocol publication, I would still have liked to have seen some further clarification on the method within this paper, both in terms of its potential limitations within the discussion, and to strengthen the reader's conviction in this method to recruit participants within the trial - perhaps by presenting recruitment trees to demonstrate heterogeneity in the sample within supplemental files? Linked to this I wondered whether seeds identified through local CBOs had social networks that extended and crossed over between women involved in transactional and commercial sex. Further detail on the recruitment process from each seed would have been useful. 

The authors present the terms commercial and transactional uncritically as categorisation that forms the basis of their analysis but raise the fact that there are issues in terms of reaching women involved in transitional sex in the discussion. It might have been helpful if they had included a clear working definition for this study within the Yaounde context. They also don't identify the potential relationality between transactional and commercial sex and the fact that these 'categories' are actually on a continuum in terms of identity, behaviour and risk exposure. Using these as distinct categories uncritically within the methodology and presentation of the results might need to be discussed within the Discussion section.

One small final point - what proportion of the sample were under 21? How feasible was it obtaining parental consent for younger women engaged in transactional or commercial sex work? Why was this approach used? I wonder if it impacted the recruitment?

Other than these minor comments and points of clarification the paper is well described, the study has been well conducted and addresses an important potential contribution to addressing risk behaviour exacerbated through health shocks, particularly for women engaged in transactional sex and should be considered for publication.

The authors might also like to consider the potential impact of the bias towards self-selection since the analysis was based only on women who returned for randomisation.

[LINK]

---

## [Decision Letter · Decision Letter 2]

29 Apr 2024

Dear Dr. Lépine,

Thank you very much for re-submitting your manuscript "The effect of protecting women against economic shocks to fight HIV in Africa: Evidence from the POWER randomized controlled study in Cameroon" (PMEDICINE-D-24-00181R2) for review by PLOS Medicine.

Thank you for your detailed response to the editors' and reviewers' comments. I have discussed the paper with my colleagues and the academic editor, and it has also been seen again by two of the original reviewers. The changes made to the paper were satisfactory to the reviewers. As such, we intend to accept the paper for publication, pending your attention to the editorial comments below in a further revision. When submitting your revised paper, please once again include a detailed point-by-point response to the editorial comments.

[LINK]

In revising the manuscript for further consideration here, please ensure you address the specific points made by each reviewer and the editors. In your rebuttal letter you should indicate your response to the reviewers' and editors' comments and the changes you have made in the manuscript. Please submit a clean version of the paper as the main article file. A version with changes marked must also be uploaded as a marked up manuscript file. Please also check the guidelines for revised papers at http://journals.plos.org/plosmedicine/s/revising-your-manuscript for any that apply to your paper. 

We ask that you submit your revision within 1 week (May 06 2024). However, if this deadline is not feasible, please contact me by email, and we can discuss a suitable alternative.

Please do not hesitate to contact me directly with any questions (aschaefer@plos.org). If you reply directly to this message, please be sure to 'Reply All' so your message comes directly to my inbox.

We look forward to receiving the revised manuscript.

Sincerely,

Alexandra Schaefer, PhD

Associate Editor 

PLOS Medicine

plosmedicine.org

Requests from Editors:

GENERAL COMMMENTS

Please check the grammar, spelling, and use of tenses throughout the main manuscript and revise the main text for clarity. When referring to tables and figures in the main manuscript, please do not write "see Table A3", but add parentheses referring to the table at the end of a sentence (e.g. (Table A3)).

FINANCIAL DISCLOUSRE

Please update the Financial Disclosure Statement in the online submission form using the information provided under "Funding," lines 508-510.

ABSTRACT

1) Please include the age of the participants.

2) Please include the recruitment dates under Methods and Findings.

3) l.13: We suggest changing the word ‘falls’ to ‘decreases’.

4) l.15: Please change to “We investigated...”

5) ll.20-22: Please change to: ““Participants were randomly assigned (1:1) to receive free health insurance for themselves and their economic dependents for 12 months either at the beginning of the study (intervention; n=XX) or at the end of the study (control; n=XX).”

6) ll.22-25: Please clearly specify the primary outcome of the trial and timing of assessment and indicate the planned population for the primary outcome. 

7) ll.25-27: Please specify the time point of this finding (Was this at endline?).

8) l.26: We suggest exchanging “treatment” with “intervention”.

9) ll.28-29: “There was also no effect on STI acquisition.” – Please clarify whether this applies for both strata.

10) l.30: Please write FSW in full.

11) l.31: Please temper claims of primacy of results by stating, "to our knowledge" or something similar (“The study provides, to our knowledge, the first evidence of the effectiveness..”).

AUTHOR SUMMARY

1) For each section, please use the questions only as subheadings (i.e. ‘Why Was This Study Done?’, ‘What Did the Researchers Do and Find?’, ‘What Do These Findings Mean?’).

2) l.57: We suggest replacing "treatment" with "intervention".

3) ll.61-63: Under ‘What Did the Researchers Do and Find?’, we suggest changing the last bullet point to: “The effect of the intervention on primary outcomes was large among women engaging in transactional sex, but there was no effect of the intervention in the commercial sex strata.”

4) ll.73-74: Under ‘What Do These Findings Mean?’, we suggest changing the last bullet point to: “The results also highlight that women in transactional sex are more affected by HIV than sex workers, a finding that warrants attention from policy makers.”

5) In the final bullet point of ‘What Do These Findings Mean?’, please describe the main limitations of the study in non-technical language.

INTRODUCTION

1) ll.98-99: Please note that there is a single upper quotation mark (“defined as ‘an unexpected”). We suggest removing the quote, the reference is sufficient.

2) ll.100-102: We suggest changing this to: “Within such relationships, women are incentivised to provide unprotected sex to men [14] because they can earn more, for example between 9% more in Kenya and 66% more in India for unprotected sex [15–17].”

3) l.104: Please change to: “…to provide basic needs [13,18].”

4) l.114: Please define “STI” at first use.

METHODS AND RESULTS

1) l.143: Please also mention that the trial protocol is available as supporting information and include the appropriate reference.

2) l.156: Please define “FSW” at first use.

3) l.160: Please define or briefly explain the term “sugar daddies”.

4) l.175: Please change “23 transactional and 31 commercial women” to “23 women in the transactional strata and 31 in the commercial strata” or similar.

5) l.181: Please define “ART” at first use.

6) l.204: We suggest exchanging “treatment” with “intervention”. Please revise throughout the main manuscript including tables and figures (including those in the Supplementary Material).

7) Table 1: Please define “Obs” and “PrEP”. Please add a definition for “sugar daddies”.

8) l.233: Please remove the footnote and include the information in the text.

9) l.238: Please define “ID” at first use.

10) ll.276.277: Please change to: “….and we collected data on unprotected sex using the double-list experiment method (See appendix 7.2 for details).”

11) ll.296-310: Please revise the two paragraphs for grammar and tense.

12) ll.313-323: Please use the past tense when describing the statistical analysis.

13) l.333 ff: Please use the past tense when writing your results (see GENERAL comments).

14) ll.340-342: Please change to: “The absence of effect could be due to serious problems found with the STI tests because they were not effective in determining STI status with certainty.”

15) ll.343-344: Please change to: “At endline, as many samples as possible were retested in the ELISA laboratory using new types of assays.”

16) l.355: Please define “ME” at first use.

17) Table 4: Please report exact p-values. Please define “OLS” and “Obs”. In the table description, please briefly explain what PPS-4 and UCLA 3 point scale are and/or provide references. 

18) l.381: Please remove any footnotes and include the information provided in the footnotes in the main text, if desired.

19) Table 5: Please report exact p-values. Please define “OLS”, “ME”, and “CI”. 

DISCUSSION

1) ll.407-408: Please change to: “Our results show the importance of including transactional sex in HIV transmission research, as we found 9 times more infections among women in transactional sex than among women in commercial sex in the control group.”

2) ll.441-444: Please do not report exact values in the Discussion, as this should be limited to the Results section (please revise throughout the Discussion). Please change to: "This may explain why there was no increase in the likelihood of visiting a professional among FSWs compared to women who engage in transactional sex."

3) l.467 and l.500: Please temper claims of primacy of results by stating, "to our knowledge" or something similar.

FIGURES

1) Figure 1: Please define “STI”. We also suggest that the figure description briefly explain what the figure is supposed to show (as described in the main text).

2) Figure 2: In the figure description, please describe the meaning of the bars and whiskers. Please describe the meaning of the green percentages in the figure, and we suggest adding the n numbers for the different groups to the description.

SUPPLEMENTARY MATERIAL

1) You have indicated that you have completed the CONSORT checklist. It appears that the completed CONSORT checklist was not uploaded as a supporting information file. When completing the checklist, please remember to use section and paragraph numbers, not page numbers.

2) We were unable to access the file "data_plos med.dta". Could you please provide the contents of the file in an alternative way?

3) Figure A1: Please include the CONSORT flowchart in the main manuscript.

4) Figure A2: Please define “STI”.

5) Figure A3: Please present numerators and denominators for percentages.

6) Table A1-A10: Please see the comments for the main tables and revise accordingly. Please report exact p-values and ensure to define all abbreviations (e.g. “Obs”, “ME”, “STI”,”OR”, etc.)

7) Table A9: What do Magenta and Teal refer to?

8) Table A10: Please remove the asterisks given you are reporting the exact p-values.

REFERENCES

1) Please revise your references. Some references, e.g. [34] and [41], appear to be under review and cannot be listed in the reference list until they have been accepted for publication or are publicly available on a preprint archive. The information may be cited in the text as a personal communication with the author if the author provides written permission to be named. Alternatively, please provide a different appropriate reference. 

2) Please revise references [4] and [7] as they do not appear to provide sufficient detail. Please check if this applies to any other references.

3) Where website addresses are cited, when specifying the date of access, please use the word “accessed” instead of “cited” (e.g. [accessed: 12/06/2023]).

SOCIAL MEDIA

To help us extend the reach of your research, please provide any X (formerly known as Twitter) handle(s) that would be appropriate to tag, including your own, your co-authors’, your institution, funder, or lab. Please enter in the submission form any handles you wish to be included when we post about this paper.

Comments from Reviewers:

Reviewer #1: The article has improved considerably. At this point, I suggest a very careful reading of the entire text, which contains some lapses, and of the references, which also have several errors (there are articles without authors).

Thank you.

Reviewer #2: Thank you to the authors for addressing all my previous comments well. I have no further issues to raise.

[LINK]

General Editorial Requests

---

## [Editor Report · Decision Letter 3]

25 Jun 2024

Dear Dr. Lépine,

Thank you very much for re-submitting your manuscript "The effect of protecting women against economic shocks to fight HIV in Africa: Evidence from the POWER randomized controlled study in Cameroon" (PMEDICINE-D-24-00181R3) for review by PLOS Medicine.

Thank you for your response to the editors' and reviewers' comments. I have discussed the paper with my colleagues and there remain a few outstanding requests which need to be carefully addressed prior to publication.

[LINK]

In revising the manuscript for further consideration here, please ensure you address the specific points made by each reviewer and the editors. In your rebuttal letter you should indicate your response to the reviewers' and editors' comments and the changes you have made in the manuscript. Please submit a clean version of the paper as the main article file. A version with changes marked must also be uploaded as a marked up manuscript file. Please also check the guidelines for revised papers at http://journals.plos.org/plosmedicine/s/revising-your-manuscript for any that apply to your paper. 

We ask that you submit your revision within 1 week (Jul 02 2024). However, if this deadline is not feasible, please contact me by email, and we can discuss a suitable alternative.

Please do not hesitate to contact me directly with any questions (atosun@plos.org). If you reply directly to this message, please be sure to 'Reply All' so your message comes directly to my inbox.

We look forward to receiving the revised manuscript.   

Sincerely,

Alexandra Tosun, PhD

Associate Editor 

PLOS Medicine

plosmedicine.org

Requests from Editors:

GENERAL COMMENTS

Please remove the terms/descriptions "midline" and "endline" throughout the paper and refer to these time points as 6 months and 12 months post-intervention (including in display items such as figures and tables).

FINANCIAL DISCLOSURE

Please revise the statement once more. The funding statement should include: specific grant numbers, initials of authors who received each award, URLs to sponsors’ websites. Also, please state whether any sponsors or funders (other than the named authors) played any role in study design, data collection and analysis, the decision to publish, or preparation of the manuscript. If they had no role in the research, include this sentence: “The funders had no role in study design, data collection and analysis, decision to publish, or preparation of the manuscript.”

ABSTRACT

1) l.18: Please remove the term “high-risk women” and use patient-centered language instead, i.e., “women at high risk of HIV infection”.

2) ll.19-20: Please check the numbers presented (1506 versus 1508), and revise throughout the main text. Please change to “Between June and August 2021, we recruited 1,508 adolescent girls and women over age 15 years who were involved in transactional sex (n=753) or commercial sex (n=755), using snowball sampling.”

3) ll.23-26, please change to: We collected data on socioeconomic characteristics of participants. Primary outcomes included incidence of HIV and sexually transmitted infections (STIs). Outcomes were measured at baseline, 6 months after randomisation, and 12 months after randomisation.

4) Please note that only the primary outcome of the trial should be reported in your Abstract. Our policy is that secondary outcomes should only be included in the Abstract if all secondary outcomes are included. For trials that have many secondary outcomes, such as yours, the abstract should be limited to reporting the primary outcome.

5) l.30: Please change “the commercial sex strata” to “women and girls involved in commercial sex” (or similar). Please revise accordingly throughout the main text.

6) l.31: Please remove the term ‘biomarkers’. Editorial suggestion: The main limitations of this study were the challenges of collecting reliable STI incidence data and the low incidence of HIV in women and girls engaged in commercial sex, which might have prevented detection of study effects.

7) Please include the details provided in lines 53-55 in the Abstract. Editorial suggestion: Participants were randomly assigned (1:1) to receive free health insurance for themselves and their economic dependents for 12 months either at the beginning of the study (intervention; n=579; commercial sex n = 289, transactional sex n = 290) from November 2021 or at the end of the study 12 months later (control; n=568; commercial sex n = 290, transactional sex n = 278).

AUTHOR SUMMARY

1) ll.51-55: Please combine the second and third bullet point under ‘What Did the Researchers Do and Find?’. 

2) ll.58-59, please change to: Among women who received the intervention, we found large reductions in HIV infection and reductions in risky sexual behaviors for women and girls engaging in transactional sex, but there was no effect in women and girls engaging in commercial sex.

3) Under ‘What Do These Findings Mean?’, please swap the last two bullet points as the main limitation(s) should be the last bullet point.

INTRODUCTION

1) l.71, please change to: “Women aged 15-24 years in Sub-Saharan Africa…”

2) l.75, please change to: “…adolescent girls aged 15–19 years..”

3) ll.78-80: Please remove the quotes. Editorial suggestion: There is growing evidence that heterosexual sex as part of commercial and transactional relationships (i.e., non-commercial sexual relationships in exchange for material support and benefits) are a key driver of the HIV epidemic in sub-Saharan Africa.

4) ll.86-89: Please provide references.

5) ll.106-107: Please rephrase to “Use of cash transfers is one type of structural intervention used to address the underlying causes of HIV risk by improving income levels.”

6) l.115: Please rephrase to “…have shown some success, including lottery-based financial incentives…”

METHODS AND RESULTS

1) l.139: Please provide an appropriate reference for trial protocol as supporting information, i.e. “The protocol paper with more details on the methods can be found in the supporting information (S2 Study protocol) [30].”

2) l.140: Please remove the sentence. The flow diagram may be referenced later (see comment l.334).

3) Figure 1: In flow diagram, for the boxes showing the follow-up section, please first show the total numbers of participants at follow-up with the numbers lost-to-follow up listed below.

4) ll.146-153: Please note that there are two identical paragraphs. Please remove one.

5) l.162: Please change to “…which we defined…”.

6) l.195: Please also reference the Supporting Information file.

7) l.200: Please reference figures/tables in parentheses, i.e. “…two months after the baseline survey had taken place in order to minimise attrition (Figure A1).”. Please change the label of the supplementary figure from ‘Figure A1’ to ‘S1 Figure’ (for further details see comment under ‘Supplementary Material’). Please revise throughout the main text.

8) l.223: Please remove the sentence “Figure 1 displays the flowchart of respondents in the analysis.”. It is sufficient to mention the flowchart once, which we suggest to do at the beginning of the Results section.

9) ll.223-225, please change to: By completing randomisation after baseline only for returning participants, we reduced the attrition rate but may have reduced the generalisability of our results.

10) Table 1: Please re-name to “Summary characteristics at baseline”.

11) l.246: Please define ‘ID’ at first use.

12) ll.253-255, please change to: “The primary outcome measure was HIV incidence, measured by rapid serological blood tests, with positive and indeterminate tests retested by enzyme-linked immunosorbent assay (ELISA), and STI incidence, including chlamydia, gonorrhoea, trichomoniasis and syphilis.”

13) l.261: Please change to ‘…per protocol..’.

14) ll.264-267: We feel this part is currently rather unclear. Please clarify the meaning of ‘intermediate’ – do you mean secondary outcomes? What does ‘intensity of commercial or transactional sex’ mean? Please list the prespecified secondary outcomes in clearly and in explicit detail and ensure that all prespecified outcomes are listed.

15) l.271, please change to: “We hypothesized…”

16) l.273, please change to: ”…lead to an increase in STIs treated,..”

17) l.277: Please write ‘OOPs’ in full.

18) l.293: Please remove the abbreviation MDE, as it is only used one additional time; you should then spell out “minimal detectable effect” at line 297.

19) l.296: Please change ‘sub-samples’ to ‘sub-groups’.

20) l.313: The term “attritors” to describe individuals lost to follow-up is not standard. Please revise to something more conventional. 

21) Figure 2: Please write ‘STI’ in full in the figure or add a definition in the figure description. Please add a brief explanation of what the figure depicts.

22) l.332, please change to: “…whether the respondent had a child.”

23) l.334: At the beginning of the Results section, please describe briefly the recruitment and enrollment schedule (i.e. participant numbers) of the trial. Please reference the flow diagram here. 

24) l.337: Please reference figures/tables in parentheses.

25) l.342: Please define ‘CI’ and ‘AOR’ at first use.

26) l.343: Please rephrase “and when the samples were pooled” to avoid referring to participants as samples. Editorial suggestion: “…and in both strata combined”

27) Figure 3: Please change the x-axis title to ‘Intervention status’ and replace the word ‘treatment’ with ‘intervention’ in the whole figure. The y-axis is labeled ‘HIV prevalence’ – do you mean HIV incidence? Also, we suggest changing the y-axis to showing percentages. In the figure description, please add details about the number of participants included in each group, the meaning of the bars and whiskers and a brief explanation of what the figures shows (e.g. the time point measured, what the intervention entails etc.). We also suggest re-naming the first two graphs ‘Commercial Sex’ and ‘Transactional Sex’. Please remember that all figures and/or tables should be self-explanatory on a stand-alone basis. Please revise accordingly and provide sufficient detail, including the title.

28) ll.364ff: Causal language - In trials, there is usually a distinction in the language in terms of causal vs associational for primary and secondary trial outcomes. Please use associational language for secondary outcomes.

29) ll.366ff: Please remove any descriptions of percentage increases/decreases (50% increase, 43% increase, 16% reduction).

30) ll.368-372: Please break down the sentence into multiple sentences to improve readability. "which was statistically significant at 10%" - please refrain from describing results as significant if the analysis was not pre-specified in the SAP. Editorial suggestion: There was an increase in condom use at last sex of about 15 percentage points as measured by the list experiment method (ME=0.152 (95% CI [-0.021,0.324]); p=0.085). We also observed a reduction in the number of sex acts by an average of 0.36 for women having transactional sex, which was not statistically significant (ME=-0.358 (95% CI [-0.781,0.065]); p=0.097).

31) l.373: We have noticed that you switch between the descriptions "women engaging in commercial sex" and "FSW". For consistency, please use the description “women and girls engaged in commercial sex” throughout the main text. Please revise accordingly.

34) Table 4: Since you provided the exact p-values in the table, please remove the asterisks from and below the table. Please define or replace the word "papy" (it does not appear that you have used the word elsewhere). In the table description, please add details about the midline and endline time points.

35) ll.384ff: Please use the past tense when describing methods and results.

36) Table 5: Since you have reported the exact p-values in the table, please remove the asterisks in and below the table. In the table description, please add details about the midline and endline time points.

DISCUSSION

1) ll.440-442, please change to: Women and girls engaged in commercial sex are considered a key population by UNAIDS [2] and have access to PrEP in Cameroon; among women and girls engaged in commercial sex in this study, 13% had ever taken PrEP at endline, while none of the women engaging in transactional sex were using PrEP at endline.

2) ll.444ff: Please remove references to tables/figures (e.g. Table 1) as you do not need to re-reference the results in the Discussion section. If you feel a specific reference to a table or appendix is necessary, please use parentheses.

3) l.471: Please avoid referring to the participants in your study as "sample" and/or "used". Please revise the main text accordingly. Editorial suggestion: The participants included in our study may be subject to bias due to the limitations of the snowball method.

SUPPLEMENTARY MATERIAL

1) Thank you for providing the SPIRIT checklist. We noticed that you did not complete the checklist by using section and paragraph numbers to indicate where the item can be found in the manuscript. Please do so by adding a "Location Reported" column. Also, in the Methods section, please add the following statement: "This study is reported as per the Standard Protocol Items: Recommendations for Interventional Trials (SPIRIT) guideline (S1 Checklist)." 

2) Tables in the appendices: Please remove the asterisks from the tables and below the tables if you are reporting exact p-values.

3) Figure A2: Please write “STI” in full.

4) Where website addresses are cited, when specifying the date of access, please use the word “accessed” instead of “cited” (e.g. [accessed: 12/06/2023]).

In the published article, supporting information files are accessed only through a hyperlink attached to the captions. For this reason, you must list captions at the end of your manuscript file. You may include a caption within the supporting information file itself, as long as that caption is also provided in the manuscript file. Do not submit a separate caption file.

When SI files are contained with a single file:

Please label the file as ‘S1 Supporting Information’.

Please apply alphabetical labelling to each table and figure contained within the S1 file. For example, ‘Fig A’ to ‘Fig Z’ and ‘Table A’ to ‘Table Z’.

Plain text does not need to be labelled and can just be given a title as necessary. For example, ‘Statistical Analysis Plan’.

Please cite tables/figures as ‘Fig A in S1 Supporting Information’ and/or ‘Table A in S1 Supporting Information’, for example.

Please cite plain text as, ‘Statistical Analysis Plan in S1 Supporting Information’, for example.

When SI files are uploaded as separate files:

Please label tables as ‘S1 Table’ (so on) and figures as ‘S1 Fig’ (and so on).

Any additional documents (protocols/analysis plans etc.) can be labelled as ‘S1 Protocol’, for example. Please cite items as exactly as labelled.

[LINK]

General Editorial Requests

---

## [Editor Report · Decision Letter 4]

31 Jul 2024

Dear Dr. Lépine,

Thank you very much for re-submitting your manuscript "The effect of protecting women against economic shocks to fight HIV in Africa: Evidence from the POWER randomized controlled study in Cameroon" (PMEDICINE-D-24-00181R4) for review by PLOS Medicine.

Thank you for your response to the editor's comments. We are looking forward to publishing your manuscript. Editorially, however, there are still minor stylistic issues that need to be addressed prior to publication. Some of these points have been addressed in your point-by-point response, but have not been changed in the manuscript. Please go over the points carefully and make sure they are addressed in the manuscript to preclude the need for further revisions. If you have any questions or concerns about these requests, please feel free to contact me at atosun@plos.org.

[LINK]

In revising the manuscript for further consideration here, please ensure you address the specific points made by each reviewer and the editors. In your rebuttal letter you should indicate your response to the reviewers' and editors' comments and the changes you have made in the manuscript. Please submit a clean version of the paper as the main article file. A version with changes marked must also be uploaded as a marked up manuscript file. Please also check the guidelines for revised papers at http://journals.plos.org/plosmedicine/s/revising-your-manuscript for any that apply to your paper. 

We ask that you submit your revision within 1 week (Aug 07). However, if this deadline is not feasible, please contact me by email, and we can discuss a suitable alternative.

Please do not hesitate to contact me directly with any questions (atosun@plos.org). If you reply directly to this message, please be sure to 'Reply All' so your message comes directly to my inbox.

We look forward to receiving the revised manuscript. 

Sincerely,

Alexandra Tosun, PhD

Associate Editor 

PLOS Medicine

plosmedicine.org

Requests from Editors:

1) Title: We suggest changing the title to “The effect of protecting women against economic shocks to fight HIV in Cameroon, Africa: the POWER randomized controlled trial”.

2) ll.20-22, please change to: Between June and August 2021, we recruited 1,508 adolescent girls and women over age 15 years who were involved in transactional sex (n=753) or commercial sex (n=755), using snowball sampling.

3) ll.22-25, please change to: Participants were randomly assigned (1:1) to receive free health insurance for themselves and their economic dependents for 12 months either at the beginning of the study (intervention; n=579; commercial sex n = 289, transactional sex n = 290) from November 2021 or at the end of the study 12 months later (control; n=568; commercial sex n = 290, transactional sex n = 278).

4) ll.26-28, please change to: Primary outcomes included incidence of HIV and sexually transmitted infections (STIs) and were measured at baseline, 6 months after randomisation, and 12 months after randomisation.

5) ll.29-31, please remove the following sentence (Per CONSORT, only the primary outcome of the trial should be reported in the Abstract): “Secondary health and sexual behaviour outcomes at baseline, 6 months after randomisation (6 months post-intervention) and 12 months after randomisation (12 months post-intervention).”

6) ll.31-34, please change to: We found that study participants who engaged in transactional sex and were assigned to the intervention group were less likely to become infected with HIV post-intervention (combined result of 6 months post-intervention or 12 months post-intervention, depending on the follow-up data available; odds ratio (OR)=0.109 (95% confidence interval (CI)[0.014,0.870]); p=0.036).”

7) Abstract ll.37-39, please change to: The main limitations of this study were the challenges of collecting reliable STI incidence data and the low incidence of HIV in women and girls involved in commercial sex, which might have prevented detection of study effects.

8) Author Summary: In the last bullet point, please spell out ‘STI’ (sexually transmitted infection).

9) ll.90-93, please change to: There is growing evidence that heterosexual sex as part of commercial and transactional relationships (i.e., non-commercial sexual relationships in exchange for material support and benefits) are a key driver of the HIV epidemic in Sub-Saharan Africa.

10) Figure 2: Please write ‘STI’ in full in the figure or add a definition in the figure description. 

11) Figure 3: For all three graphs, please:

a) change the x-axis title to ‘Intervention status’ and replace the word ‘treatment’ with ‘intervention’ below the second bar;

b) change the y-axis label to ‘HIV incidence’;

c) re-name the first two graphs ‘Commercial Sex’ and ‘Transactional Sex’

d) re-name the figure ‘Effect of free health insurance on HIV incidence’

12) Table 4: Please remove the description of the significance levels below the table.

13) Table 5: Please remove the terms ‘midline’ and ‘endline’ and write ‘6 months post intervention’ and ’12 months post intervention’ only.

14) Thank you for providing your CONSORT checklist. Please replace the page numbers with paragraph numbers per section (e.g. "Methods, paragraph 1"), since the page numbers of the final published paper may be different from the page numbers in the current manuscript.

15) Please add the following statement, or similar, to the Methods (for example line 160): "This study is reported as per the Consolidated Standards of Reporting Trials (CONSORT) guideline for randomised controlled trials (S1 Checklist)."

16) ll.431ff: Thank you for your response regarding causal versus associational language. Typically, trials are only powered for the primary outcome(s), which justifies the use of causal language. Unless you can demonstrate that the study was also powered for the secondary outcomes, we ask that you remove any causal language and instead use associational language when reporting the secondary outcomes. At the very least, please include a qualifier (e.g., "suggesting a potential causal link...") to mitigate the causal relationship.

[LINK]

General Editorial Requests

---

## [Editor Report · Decision Letter 5]

28 Aug 2024

Dear Dr Lépine, 

On behalf of my colleagues and the Academic Editor, Ruanne V. Barnabas, I am pleased to inform you that we have agreed to publish your manuscript "The effect of protecting women against economic shocks to fight HIV in Cameroon, Africa: the POWER randomized controlled trial" (PMEDICINE-D-24-00181R5) in PLOS Medicine.

I appreciate your thorough responses to reviewers' and editors' comments, and your engagement throughout the editorial process. We look forward to publishing your manuscript, and editorially there is only one remaining minor presentation point that should be addressed prior to publication. We will carefully check whether this change has been made. If you have any questions or concerns regarding the final request, please feel free to contact me at atosun@plos.org.

Please see below the minor point that we request you respond to:

1) Figure 3: For all three graphs, please change the y-axis labels to 'HIV incidence'.

Before your manuscript can be formally accepted you will need to complete some formatting changes, which you will receive in a follow up email (including the editorial point above). Please be aware that it may take several days for you to receive this email; during this time no action is required by you. Once you have received these formatting requests, please note that your manuscript will not be scheduled for publication until you have made the required changes.

PRESS

Sincerely, 

Alexandra Tosun, PhD 

Associate Editor 

PLOS Medicine